# SLM-MUX: Orchestrating Small Language Models for Reasoning

**Chenyu Wang**[1*†]    **Zishen Wan**[1,2*†]    **Hao Kang**[2]    **Emma Chen**[1]
**Zhiqiang Xie**[3]    **Tushar Krishna**[2]    **Vijay Janapa Reddi**[1]    **Yilun Du**[1]

[1]Harvard University    [2]Georgia Institute of Technology    [3]Stanford University

## Abstract

With the rapid development of language models, the number of small language models (SLMs) has grown significantly. Although they do not achieve state-of-the-art accuracy, they are more efficient and often excel at specific tasks. This raises a natural question: can multiple SLMs be orchestrated into a system where each contributes effectively, achieving higher accuracy than any individual model? Existing orchestration methods have primarily targeted frontier models (e.g., GPT-4) and perform suboptimally when applied to SLMs. To address this gap, we propose a three-stage approach for orchestrating SLMs. First, we introduce SLM-MUX, a multi-model architecture that effectively coordinates multiple SLMs. Building on this, we develop two optimization strategies: (i) a model selection search that identifies the most complementary SLMs from a given pool, and (ii) test-time scaling tailored to SLM-MUX. Our approach delivers strong results: Compared to existing orchestration methods, our approach achieves up to 13.4% improvement on MATH, 8.8% on GPQA, and 7.0% on GSM8K. With just two SLMs, SLM-MUX outperforms Qwen 2.5 72B on GPQA and GSM8K, and matches its performance on MATH. We further provide theoretical analyses to substantiate the advantages of our method. Additional experiments show that the core principle of SLM-MUX extends to open-ended generation tasks (e.g., HumanEval) and benefits other model classes, including frontier LLMs and domain-specific fine-tuned SLMs. In summary, we demonstrate that SLMs can be effectively orchestrated into more accurate and efficient systems through the proposed approach.

## 1 Introduction

Recent years have witnessed a surge of small-sized language models (SLMs) containing billions to tens of billions of parameters (Wang et al., 2024a; Javaheripi & Bubeck, 2023; Guo et al., 2025; Allal et al., 2025). While these models may underperform state-of-the-art frontier language models, which usually contain hundreds of billions to trillions of parameters, on any given query, they offer substantially lower inference costs, are more affordable to train and fine-tune, and allow edge deployment due to their small size (Belcak et al., 2025). Meanwhile, frontier models have reached trillion-parameter scales where further increases in size and training data yield diminishing returns. This mirrors a well-known challenge in computer architecture two decades ago: when enlarging single CPU cores no longer delivered proportional performance gains, computer architects turned to designing multi-core processors, where multiple smaller cores working together enabled sustained improvements. This parallel suggests that combining multiple SLMs could offer a promising alternative to scaling ever-larger frontier models.

Recent works have explored orchestrating multiple LLMs (e.g., GPT-3.5 and GPT-4o), combining them into one system to process an input collaboratively. Representative approaches include Mixture-of-Agent (Wang et al., 2024b), LLM-Debate (Du et al., 2023), and Multi-Agent Verification (Lifshitz et al., 2025). These approaches share a key assumption: that models possess strong reasoning and deliberation abilities, so that interaction through natural language can reliably correct mistakes. However, when applied to SLMs, this assumption no longer holds. Our study finds that *such discussion-based orchestration often fails to improve performance for SLMs*, and in some cases even

---

[*]Equal contribution.

[†]Correspondence to `chenyu_wang@seas.harvard.edu`, `zishenwan@seas.harvard.edu`.

reduces accuracy by over 5%. Instead of correcting mistakes, SLMs tend to fall into groupthink during interaction, amplifying errors rather than mitigating them. The assumptions that language models can correct each other's answers behind existing orchestration methods do not hold for SLMs (Taubenfeld et al., 2024; Huang et al., 2024; Liu et al., 2023; Fu et al., 2025).

To address this issue, we propose **SLM-MUX**, a multi-model architecture for effectively orchestrating SLMs while avoiding explicit text exchanges between models. Our key insight is that SLM-MUX leverages complementary abilities from different models by selecting outputs based on confidence scores without any model training.

After introducing SLM-MUX, another question arises: which models should be orchestrated together? Not all combinations are effective – if one model is weaker across all dimensions, it provides no benefit when paired with a stronger one. In contrast, combining models with complementary strengths (e.g., one stronger in algebra, another in geometry) allows the system to succeed where a single model would fail.

Figure 1: **Head-to-Head Comparison of SLM-MUX with Other Methods.** SLM-MUX outperforms existing methods such as Self-Consistency (SC) (Wang et al., 2023), Mixture-of-Agents (MoA) (Wang et al., 2024b), LLM-Debate (Du et al., 2023), Multi-Agent Verification (MAV) (Lifshitz et al., 2025), and Agent Forest (Li et al., 2024). Results reported on MATH dataset with SLMs.

To address this, we develop a **model selection search strategy** for SLM-MUX, which systematically evaluates and identifies model subsets with complementary strengths. By maximizing union accuracy while penalizing overconfident contradictions, the search procedure finds the most suitable models for a given model budget.

In addition, we explore **compute scaling strategies** for the selected model ensembles to further enhance performance. By adjusting the number of models and samples at inference time, we further boost performance and identify practical sweet spots in the accuracy-compute tradeoff.

Our experiments demonstrate significant improvements across multiple benchmarks. By combining only two SLMs, we achieve accuracy improvements of up to 6.7% on MATH, 5.7% on GPQA, and 4.8% on GSM8K, compared to the best-performing single SLMs in the system. Our method consistently outperforms existing discussion-based approaches for SLMs, with gains of up to 13.4% on MATH, 8.8% on GPQA, and 7.0% on GSM8K. Most importantly, with just two SLMs, SLM-MUX outperforms Qwen 2.5 72B on GPQA and GSM, and matches its performance on MATH.

Finally, we complement these empirical findings with theoretical and experimental analyses. Our approach shows superiority in multiple scenarios compared with previous methods (Figure 1).

Our main contributions are as follows: **(i) We identify a fundamental limitation of existing orchestration methods:** Through systematic evaluation, we demonstrate that existing discussion-based methods, which show consistent improvements for frontier LLMs, actually harm performance when applied to SLMs. This counterintuitive finding challenges the assumption that orchestration methods transfer across model scales and reveals the need for SLM-specific method. **(ii) We propose SLM-MUX**, a novel multi-model architecture designed specifically for SLMs that avoids the error amplification problems of discussion-based methods. SLM-MUX achieves consistent gains across multiple benchmarks (MATH, GPQA, GSM8K) and significantly outperforms existing discussion-based methods by large margins (up to 11.6% on MATH). **(iii) We develop principled optimization strategies** for the SLM-MUX, including model selection search that identifies complementary model selections and compute scaling strategies, further boosting performance while maintaining efficiency.

## 2  RELATED WORK

**Discussion-based Orchestration Methods**. We use discussion-based orchestration to refer to orchestration schemes where multiple LM instances exchange or evaluate natural-language messages (Fu et al., 2025)—such as proposing answers, critiquing or debating, verifying from different aspects, and finally aggregating into one output. Representative approaches include Mixture-of-Agents (Wang et al., 2024b), which uses a dedicated LLM to aggregate outputs from several models; LLM-Debate (Du et al., 2023), where models critique and refine each other's reasoning; and Multi-

Figure 2: **Comparing SLM-MUX (Ours) with Existing LLM Orchestration Methods.** (a) Mixture-of-Agents, (b) LLM-Debate, (c) Multi-Agent Verification, (d) SLM-MUX (Ours).

Agent Verification (Lifshitz et al., 2025), which assigns models to independently evaluate candidate solutions before selecting the final answer. These methods assume that participating models have sufficient reasoning ability to self-correct through interaction. Prior evaluations have been conducted on frontier LLMs, while their effectiveness for SLMs remains unstudied.

**Optimization for Multi-LM Orchestration**. Given these orchestration methods, some works study how to further improve their performance—e.g., how to select models to include, how to optimize prompts, or how to adapt the architecture for specific tasks (Chen et al., 2023a; Ong et al., 2025; Chen et al., 2024). Prompt and workflow optimization methods (Khattab et al., 2023; Opsahl-Ong et al., 2024; Saad-Falcon et al., 2025; Zhang et al., 2025a) generally assume strong instruction-following ability, which makes them less effective for smaller models with limited such capabilities..

Another line of work is model selection for orchestration (Chen et al., 2025; Poon et al., 2025). These methods often select models based on accuracy, assuming that combining models with higher standalone accuracy will yield stronger orchestrations. However, most selection criteria are not end-to-end: they evaluate models independently without directly assessing the performance of the overall orchestration. This overlooks how models interact with each other—overconfident but incorrect predictions from one model can dominate and suppress correct predictions from others, meaning that the best standalone models may not yield the best orchestration.

**Test-time Scaling Strategies**. Test-time scaling methods improve performance by using additional computation during inference without retraining (Snell et al., 2024; Muennighoff et al., 2025; Zhang et al., 2025b). A common single-model approach is self-consistency (Trad & Chehab, 2025; Thirukovalluru et al., 2024; Chow et al., 2024), which draws multiple samples from one model and selects the majority answer; accuracy typically improves as the number of samples increases. Agent Forest (Li et al., 2024) extends this idea to multiple models by collecting one output from each model and applying majority voting across all answers.

## 3 METHODS

In this work, we set out to ask two critical questions: given a pool of available SLMs, how can we (i) orchestrate their outputs to achieve the best overall performance, and (ii) select an effective subset of models that maximizes accuracy?

To answer question (i), we present the SLM-MUX (Section 3.1), a simple yet effective orchestration method. To answer question (ii), we propose model selection search (Section 3.2) that identifies complementary subsets from dozens of available SLMs. Finally, we explore compute scaling strategies (Section 3.3) to further enhance the reasoning accuracy during inference.

### 3.1 SLM-MUX FOR ORCHESTRATING MULTIPLE SMALL LANGUAGE MODELS

At a high level, our intuition is that we do not need to let SLMs discuss with each other. Instead, we can develop a simple rule-based method that estimates the confidence of each model's answer and then selects the final output from the model with the highest confidence. We term our method **SLM-MUX**, which operates in two phases.

**Independent Generation Phase.** For a given question, we first let each SLM independently generate multiple candidate responses to the same query prompt with temperature $> 0$, producing a pool of sampled answers per model.

**Question:** Which of the following physical theories never requires regularization at high energies?
(A) Superstring Theory
(B) Classical Electrodynamics
(C) Quantum Electrodynamics (QED)
(D) Quantum Chromodynamics (QCD)
**Correct Answer: (A)**

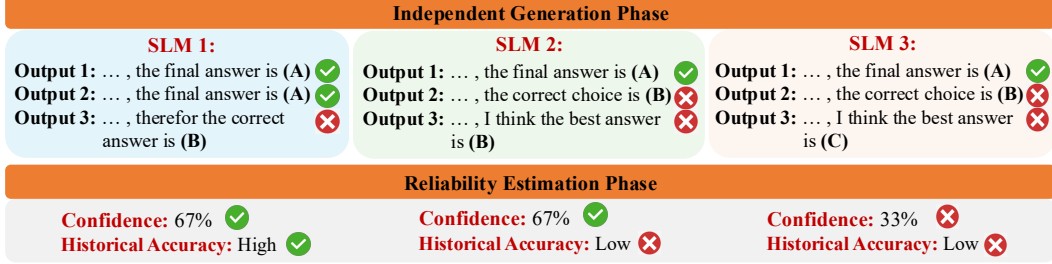

Figure 3: **Illustration of SLM-MUX Workflow.** (1) Each SLM first independently generates multiple outputs for the same question. (2) The most frequent answer from each SLM is selected, and its frequency in the answer pool is used as the confidence score. (3) The answers with the highest confidence score are selected. (4) If multiple answers share the same confidence score, the tie is broken by selecting the answer from the SLM with the highest accuracy on the validation set.

---

**Algorithm 1** SLM-MUX Working Flow

---

**Input**: Models $M_1, \ldots, M_n$, query $x$, samples per model $k$, validation accuracies $a_1, \ldots, a_n$
**Output**: Final answer $\hat{y}$

    *Independent Generation: each model produces multiple candidate answers independently*
1: **for** $i = 1, \ldots, n$ **do**
2:     Sample $k$ answers $Y_i = \{y_i^{(1)}, \ldots, y_i^{(k)}\}$ from $M_i$
3:     Compute $f_i(y) = \frac{1}{k} \sum_{j=1}^{k} \mathbf{1}\left(y_i^{(j)} = y\right)$
4:     Let $y_i^* = \arg\max_y f_i(y)$ and set $s_i = f_i(y_i^*)$
    *Confidence Estimation: measure confidence and break ties by validation accuracy*
5: $S_{\max} = \max_i s_i, \quad I^* = \{i \mid s_i = S_{\max}\}$
6: **if** $|I^*| = 1$ **then**
7:     $i^* \leftarrow$ the unique index in $I^*$
8: **else**
9:     $i^* \leftarrow \arg\max_{i \in I^*} a_i$
10: **return** $\hat{y} = y_{i^*}^*$

---

**Confidence Estimation Phase.** We evaluate the confidence of each SLM's outputs by measuring their consistency across their own outputs. Intuitively, a model that places higher probability mass on the correct answer will reproduce equivalent answer across samples, whereas an uncertain model will produce varied outputs. For instance, if SLM A produces three equivalent answers while model B produces three different ones, the answers from model A are more consistent and should be selected. This correlation between consistency and correctness is observed by previous papers. (Wang et al., 2023; Xie et al., 2024; Taubenfeld et al., 2025; Chen et al., 2023b), and we empirically revalidate this observation in Appendix D.1.

In cases where two SLMs are equally consistent but disagree, we use their validation accuracy as a tie-breaker. Prior work has shown that consistency is strongly correlated with correctness, which provides a rationale for this design.

For more details, Algorithm 1 summarizes the workflow step by step. Figure 3 provides a visual example of the workflow. The evaluation of SLM-MUX is presented in Section 4.2.

## 3.2 MODEL SELECTION SEARCH FOR SLM-MUX OPTIMIZATION

At a high level, the idea of model selection search is to combine models with complementary skills. The goal is not simply to add more models, but to bring new capabilities as we add models. Figure 4 illustrates this intuition: Qwen2.5-7B consistently outperforms Llama3.2-3B across all subjects, so combining them offers no capability beyond what Qwen2.5-7B already provides. In contrast, Mistral

Small 24B and Qwen2.5-7B Mistral Small 24B and Qwen2.5-7B excels in different subjects, making their combination more effective than either model individually.

We frame model selection as a search problem on the validation set with two competing objectives. Our first objective is **Union Accuracy**, which reflects the overall accuracy of the system. The higher the union accuracy is, the more questions a system can potentially answer. Formally, let $\mathcal{M} = \{m_1, \dots, m_K\}$ denote the set of candidate models and $\mathcal{D}$ the validation set. For each model $m_i \in \mathcal{M}$, we record the subset of validation instances it solves correctly. Given a candidate subset $S \subseteq \mathcal{M}$, the union accuracy is defined as

$$\text{UnionAcc}(S) = \frac{1}{|\mathcal{D}|} \sum_{x \in \mathcal{D}} \mathbf{1}\{\exists\, m \in S\,:\, m(x) \text{ is correct}\}$$

The second objective is the **Contradiction Penalty**. It captures problematic cases where overconfident wrong answers suppress correct predictions from other models. Consider two SLMs answering the same multiple-choice question three times: the first model consistently outputs "A" (correct), while the second consistently outputs "B" (incorrect but confident). Since SLM-MUX selects based on consistency, both models would appear equally confident, making it impossible to distinguish the correct answer from the confident but wrong one. We define this penalty as the percentage of questions where at least one model consistently gives the wrong answer while another provides the correct answer:

$$\text{Contradiction}(S) = \frac{1}{|\mathcal{D}|} \sum_{x \in \mathcal{D}} \mathbf{1}\left\{ \begin{array}{l} \exists\, m_1 \in S\,:\; m_1(x) \text{ consistently wrong,} \\ \exists\, m_2 \in S\,:\; m_2(x) \text{ correct} \end{array} \right\}$$

The final objective balances these competing factors:

$$\mathcal{O}(S) \;=\; UnionAcc(S) \;-\; \lambda \cdot Contradiction(S),$$

Where $\lambda$ is a hyperparameter. Since the number of candidate models is not very large, we perform an exhaustive search. We present visualization of the two search objectives and evaluation of the searched model selection in Section 4.3.

The rationale behind this search objective is as follows: UnionAcc represents an optimistic upper bound for SLM-MUX performance. It assumes an ideal selection mechanism capable of identifying the correct answer whenever at least one model provides it, which is unrealistic in practice. Conversely, when $\lambda = 1$, the search objective represents a pessimistic lower bound of SLM-MUX accuracy. This setting assumes that in cases involving confidently wrong answers, the system will invariably select the incorrect one. In practice, due to factors such as tie-breaking rules and the presence of confidently correct answers, such a worst-case scenario will not always happen. Consequently, by employing the objective $\mathcal{O}(S) = \text{Acc}(S) - \lambda \cdot \text{Contradiction}(S)$, we estimate an approximate accuracy between the theoretical upper and lower bounds of the SLM-MUX accuracy.

### 3.3 COMPUTE SCALING STRATEGIES

Next, we empirically investigate two dimensions of test-time scaling to further enhance the performance of our SLM-MUX with selected models.

**Adding More Participating Model Types:** As we scale the model participating model types used in the system by adding more SLMs with complementary strengths, we expect the overall accuracy to improve. For each budgeted number of models, we use the search method proposed in Section 3.2 to identify the best selection from the pool.

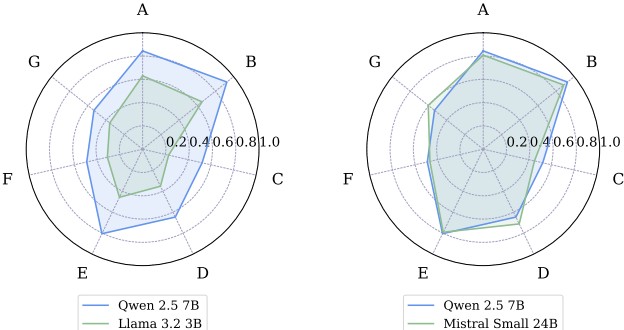

Figure 4: **Comparison of Model Choices**. Accuracy on 7 subjects for two model selection settings on MATH dataset. Subjects are denoted as: A = Prealgebra, B = Algebra, C = Intermediate Algebra, D = Number Theory, E = Counting & Probability, F = Geometry, G = Precalculus.

**Drawing More Samples per Model:** For a fixed model selection, we can increase the compute budget by scaling the number of samples drawn by each model. Since confidence is evaluated by counting the frequency of majority answers, adding more samples per model is expected to provide a more accurate confidence estimate.

These two compute scaling dimensions are evaluated in Section 4.4.

# 4 EXPERIMENTS

In our experiments, we first demonstrate the fundamental limitations of existing discussion-based orchestration methods when applied to SLMs (Section 4.1). We then evaluate the proposed SLM-MUX in Section 4.2. In Section 4.3, we assess our proposed search strategy. Finally, in Section 4.4, we examine the compute scaling strategies.

## 4.1 EXISTING DISCUSSION-BASED ORCHESTRATION METHODS HARM SLM PERFORMANCE

To understand whether orchestration methods developed for frontier LLMs are suitable for SLMs, we conduct a systematic comparison across model scales. We evaluate three prominent discussion-based methods—LLM-Debate (Du et al., 2023), Mixture-of-Agents (Wang et al., 2024b), and Multi-Agent Verification (Lifshitz et al., 2025) —using identical experimental settings on both SLMs (Llama 3.1 8B (Jiang et al., 2024), Mistral 8×7B (Grattafiori et al., 2024), Gemma 2 27B) and frontier LLMs (DeepSeek V3 (DeepSeek-AI et al., 2025), Gemini 2.0 Flash (Google Cloud, 2025), GPT-4o (OpenAI et al., 2024)). Evaluation is conducted on MATH and GPQA datasets using original code and prompts.

**Results**. As shown in Figure 5, discussion-based methods generally outperform the single best-performing models in the frontier LLM group, achieving up to a 2% increase in accuracy. However, when applied to SLMs, these discussion-based methods fail to outperform the best single model in the orchestration, and even incur accuracy drops of up to 5.5%. This performance gap is observed across all three methods and both benchmarks.

To understand this counterintuitive result, we analyze SLM behavior in discussion settings. We find that discussion-based methods amplify rather than correct errors in SLMs due to a key limitation: SLMs tend to exhibit groupthink, reinforcing incorrect reasoning during discussions rather than correcting mistakes. In Appendix C, we provide detailed analysis showing that 59.5% of failures are attributed to groupthink, and that the performance gap persists even after extensive prompt optimization.

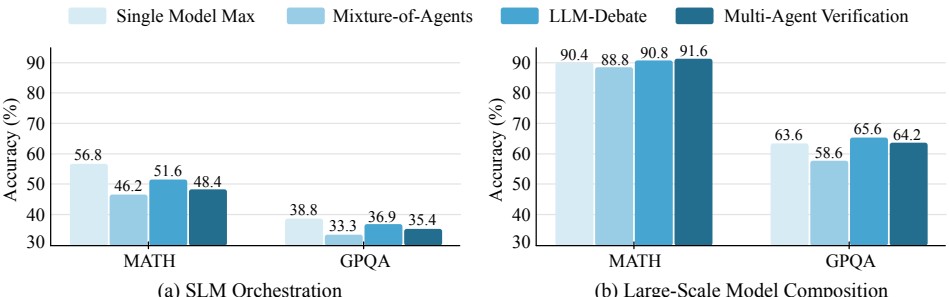

Figure 5: **Comparison of discussion-based orchestration when invoking SLMs and LLMs.** We compare three orchestration methods (Mixture-of-Agents, LLM-Debate, and Verification) using (a) SLMs (Llama 3.1 8B, Mistral 8×7B, Gemma 2 27B) and (b) frontier LLMs (DeepSeek V3, Gemini 2.0 Flash, GPT-4o) on the MATH and GPQA datasets. The baseline (*Single-Model Max*) reflects the best performance of individual models. An orchestration is considered successful if it surpasses Single-Model Max. All discussion-based methods are evaluated with temperature=0. The standard deviations of the accuracies are presented in Appendix B.3.

## 4.2 SLM-MUX ACHIEVES SLM ORCHESTRATION WHERE EXISTING METHODS FAIL

To evaluate whether our proposed SLM-MUX can successfully orchestrate SLMs, we test it against the same baselines from Section 4.1. We use Mistral 8×7B, LLaMA 3.1 8B, and Gemma 2 27B (Team et al., 2024) as base models. We implement the SLM-MUX as follows. First, we generate three rounds of answers with a temperature of 0.3. Next, we compute a confidence score by counting how often the most common answer appears across these rounds. The final answer for each model is chosen as the most frequent one; in the case of a tie, we select the answer from the model with the highest validation accuracy.

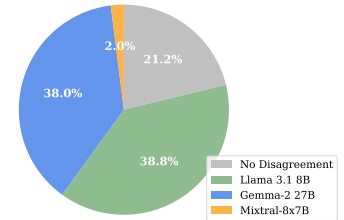

Figure 6: **Final Output Attribution**. We report the percentage of outputs contributed by each model on the MATH dataset for our SLM-MUX. These results are from the same run as in Table 1.

We evaluate three types of baselines. First, we measure
the accuracies of individual models and report the best-
performing ones. Second, we apply self-consistency to each of the three base models independently,
reporting the best-performing result as the *Single-Best-SC* baseline. Next, for comparison with exist-
ing discussion-based methods, we include LLM-Debate (Du et al., 2023), Mixture-of-Agents (Wang
et al., 2024b), and Multi-Agent Verification (Lifshitz et al., 2025). We follow the original code
and prompts described in their papers. Experiments are conducted on three benchmark datasets:
MATH (Hendrycks et al., 2021), GPQA (Rein et al., 2023), and GSM8K (Cobbe et al., 2021).

**Results**. Table 1 summarizes the results. In our experiments, we find that for SLMs, existing
orchestration methods do not consistently outperform the strongest individual base models or
self-consistency approaches. In contrast, our SLM-MUX generally achieves an accuracy gain.
Compared with other approaches, our method yields up to a 13.4% improvement on MATH, up to
8.8% on GPQA, and up to 7.0% on GSM8K. These results demonstrate that the SLM-MUX itself
provides a clear advantage over alternative orchestration approaches at the architectural level.

To better illustrate our proposed SLM-MUX, we plot the output attribution for the MATH experiment
(Table 1) in Figure 6. By selecting diverse outputs from the generation, SLM-MUX leverages the
complementary strengths of different SLMs.

| Method | MATH Acc (%) | GPQA Acc (%) | GSM8K Acc (%) |
|---|---|---|---|
| Mixture-of-Agents | $51.4 \pm 2.2$ | $33.3 \pm 3.4$ | $81.6 \pm 1.7$ |
| LLM-Debate | $51.6 \pm 2.2$ | $36.8 \pm 3.4$ | $80.8 \pm 1.8$ |
| Multi-Agent Verification | $48.4 \pm 2.2$ | $35.3 \pm 3.4$ | $86.4 \pm 1.5$ |
| **SLM-MUX (Ours)** | $\mathbf{61.8 \pm 1.2}$ | $42.1 \pm 0.3$ | $\mathbf{87.8 \pm 0.6}$ |
| Single-Best | $56.8 \pm 2.2$ | $38.9 \pm 3.5$ | $84.2 \pm 1.6$ |
| Single-Best-SC | $58.0 \pm 2.2$ | $\mathbf{42.4 \pm 3.5}$ | $86.8 \pm 1.5$ |

Table 1: **Quantitative Results.** Accuracy and standard error across MATH, GPQA, and GSM8K. "SC" denotes
self-consistency decoding (majority vote over samples from a single model), and "Single-Best-SC" reports
the highest accuracy among the three base models when each applies self-consistency individually.

### 4.3 MODEL SELECTION SEARCH BOOSTS SLM-MUX PERFORMANCE

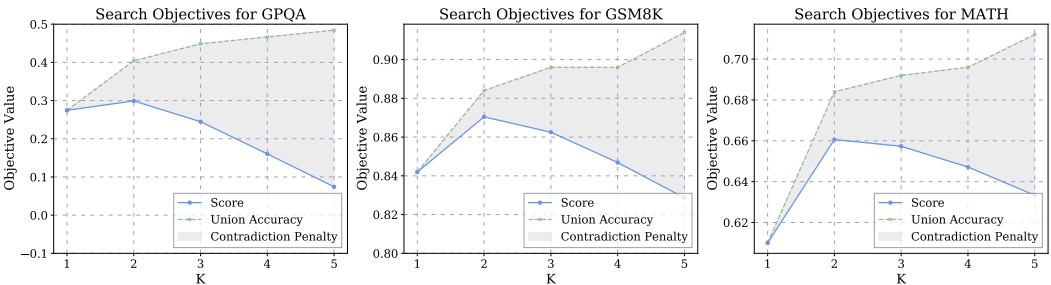

Figure 7: **Union Accuracy and Contradiction Penalty both Increases as more models are added**. We plot
the search objectives as the number of models (K) increases from 2 to 5 across three benchmarks. The green line
denotes the union accuracy across models, the grey area indicates the contradiction penalty, and the blue line
represents the overall search objective score. For each value of $K$, the plotted quantities are computed for the
single model combination that maximizes our model selection objective defined in Section 3.2.

To examine whether model selection search benefits SLM-MUX, we construct a validation set
of 500 questions sampled from the training splits of MATH, GPQA, and GSM8K. The candidate
pool consists of five SLMs: Gemma 2 27B, Llama 3.1 8B, Mistral Small 24B (Mistral AI, 2025),
Mixtral 8×7B, and Qwen2.5 7B (Qwen et al., 2025). For each question, we collect three independent
generations per model with temperature 0.5, repeating this process three times to obtain stable
accuracy estimates. The search procedure considers orchestrations with $K = 2$ to 5 models and is
guided by an objective function mentioned in Section 3, with hyperparameter $\lambda = 1$. The behavior of
this objective is illustrated in Figure 7, showing the trade-off as $K$ increases. For simplicity, we select
two representative two-model combinations from the search results for evaluation on the test set.

**Results**. Table 2 summarizes the outcome of the search. The table lists the top-performing two-model
combinations identified on the validation set, along with their evaluation on the held-out test set.
Across benchmarks, these optimized orchestrations yield consistent improvements over the strongest
individual models: accuracy increases by 4.5% on MATH, 4.4% on GPQA, and 4.3% on GSM8K.

| Benchmark | Group | Model Selection | Best Single (Acc. %) | Composed (Acc. %) | Δ (Gain) |
|-----------|-------|-----------------|----------------------|-------------------|----------|
| **MATH** | 1 | Mistral Small 24B Qwen2.5 7B | $75.5 \pm 1.5$ | $80.0 \pm 0.7$ | +4.5 |
| | 2 | Qwen2.5 7B Llama 3.1 8B | $75.5 \pm 1.5$ | $77.7 \pm 0.7$ | +2.2 |
| **GPQA** | 1 | Gemma 2 27B Mistral Small 24B | $45.1 \pm 2.8$ | $49.5 \pm 1.8$ | +4.4 |
| | 2 | Llama 3.1 8B Mistral Small 24B | $45.1 \pm 2.8$ | $48.8 \pm 0.8$ | +3.6 |
| **GSM8K** | 1 | Mistral Small 24B Qwen2.5 7B | $88.5 \pm 0.7$ | $92.8 \pm 0.6$ | +4.3 |
| | 2 | Llama 3.1 8B Mixtral 8×7B | $80.8 \pm 2.1$ | $85.2 \pm 0.7$ | +4.4 |

Table 2: **Model Selection Search and Evaluation Results.** We show the top two model groups identified by our search for each benchmark. For each group, we report the accuracy of the best-performing single model within the orchestration, the accuracy achieved by our SLM-MUX, and the resulting performance gain.

This contrasts with Section 4.2, where naive three-model combinations provide little to no benefit on GPQA. Figure 7 further illustrates the underlying trade-off: while union accuracy rises with additional models, the contradiction penalty also grows, emphasizing that effective orchestration requires balancing these competing factors rather than simply enlarging the orchestration size. In Appendix D.3, we show that the SLM-MUX architecture itself yields consistent gains even with randomly selected model combinations; the search procedure provides an effective and data-efficient way to further boost accuracy.

## 4.4 Compute Scaling Strategies Reveal Optimal Resource Allocation

To evaluate the "Adding More Participating Model Types" dimension of compute scaling, we assess how performance changes as the number of models in the orchestration increases. For each number of models from 2 to 5, we first apply the search method from Section 3.2 to identify the optimal model selection from our pool. We then evaluate SLM-MUX with selected models on the validation set. Figure 9 plots the resulting mean accuracy (blue line, left y-axis) for each value of K. To illustrate the theoretical performance ceiling of each ensemble, we also plot the union accuracy (grey line, right y-axis), defined as the percentage of questions solved by at least one model in the group. For each value of $K$ in Figure 9, we show the single model combination that achieves the highest value of our model selection objective from Section 3.2; the search procedure is used to find the best combination under a fixed $K$, rather than to choose $K$ itself.

| Benchmark | Samples | SLM-MUX | Agent Forest | Δ (Gain) |
|-----------|---------|---------|--------------|----------|
| MATH | 2 | $76.8 \pm 0.7$ | $72.3 \pm 1.5$ | +4.5 |
| | Best | $79.5 \pm 0.4$ | $79.2 \pm 0.4$ | +0.3 |
| GPQA | 2 | $46.3 \pm 2.3$ | $40.4 \pm 2.3$ | +5.9 |
| | Best | $48.8 \pm 1.2$ | $47.6 \pm 1.4$ | +1.2 |
| GSM8K | 2 | $82.1 \pm 0.7$ | $77.7 \pm 0.2$ | +4.4 |
| | Best | $86.5 \pm 0.8$ | $84.3 \pm 0.8$ | +2.2 |

Table 3: **Comparison of SLM-MUX and Agent Forest.** We compare SLM-MUX and Agent Forest in two settings: **(1)** with 2 samples per model (Samples=2), and **(2)** using the best accuracy found during scaling for each method (Samples=best). In the second setting, the number of samples per model may vary.

For the "Drawing More Samples per Model" dimension, we reuse the two groups of models listed in Table 2. We vary the number of samples per model from 2 to 9 and report the mean accuracy of SLM-MUX over three runs for each sample budget. The results are presented in Figure 8, along with a baseline, Agent Forest (Li et al., 2024), for comparison. To ensure fairness, Agent Forest is reproduced using the same models from Group 2. We report the best accuracy achieved by the SLM-MUX when scaling with Samples per Model and compare it to the accuracy of the single best model in the orchestration, as shown in Table 2.

**Results**. The effect of "Adding More Participating Model Types" varies substantially across benchmarks. On GPQA, accuracy peaks when combining two models and declines thereafter. On GSM8K, accuracy quickly saturates at two models without further gains. In contrast, on MATH, accuracy

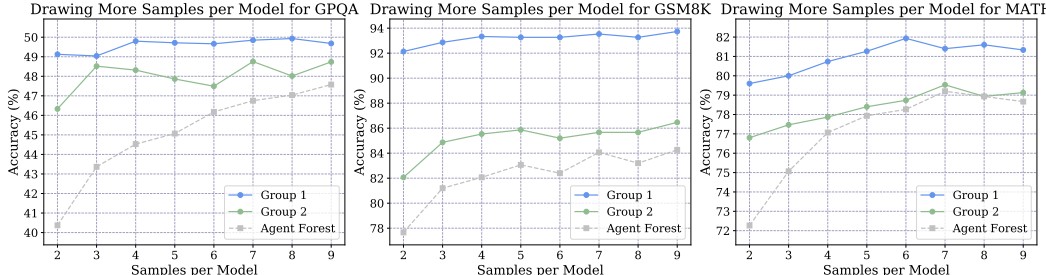

Figure 8: **Drawing More Samples per Model Improves Accuracy**. We report mean accuracy of SLM-MUX as the number of samples per model increases from 2 to 9 across three benchmarks. Group 1 and Group 2 are from Table 2. We also plot the mean accuracy of Agent Forest (Li et al., 2024) in grey line.

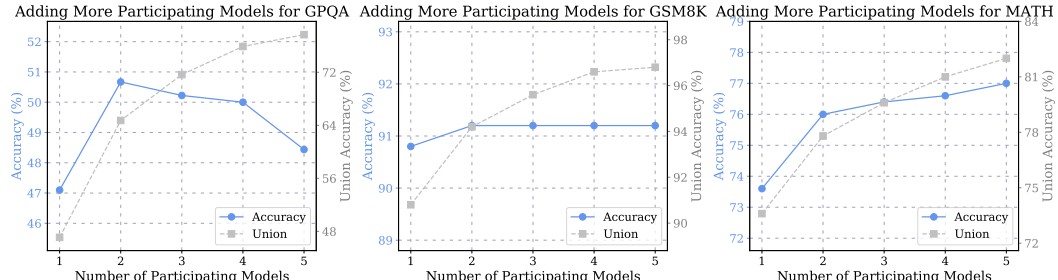

Figure 9: **Adding More Participating Models Affects Accuracy Differently**. We report the mean accuracy (blue line) of the optimal SLM-MUX obtained when using 2 to 5 models across three benchmarks. We also report the union accuracy (grey line), defined in Section 3.2. The blue line (Mean Accuracy) is plotted against the left-hand Y-axis. The grey line (Union Accuracy) is plotted against the right-hand Y-axis. For each $K$, both curves correspond to the single model combination that maximizes our model selection objective (Section 3.2) under that fixed $K$.

continues to improve as additional models are included. Despite these differences, the union accuracy of model orchestration consistently increases with more models, emphasizing the role of output contradictions among models, as elaborated in Section 3.2. We also validate this scaling behavior on the test set; see Appendix D.4 for details.

"Drawing More Samples per Model" yields more consistent improvements across benchmarks. Moreover, under this setting, our SLM-MUX systematically outperforms Agent Forest, with the largest margin observed on GPQA, where single-model accuracy is lowest.

| Benchmark | Group 1 | | Group 2 | | Qwen-2.5 72B Acc. % |
|---|---|---|---|---|---|
| | Acc. % | $\Delta$ (Gain) | Acc. % | $\Delta$ (Gain) | |
| MATH | $81.9 \pm 0.2$ | $+6.4$ | $79.5 \pm 0.4$ | $+4.0$ | $82.3 \pm 0.5$ |
| GPQA | $49.9 \pm 1.8$ | $+4.8$ | $48.7 \pm 1.2$ | $+3.6$ | $44.9 \pm 0.5$ |
| GSM8K | $93.7 \pm 0.2$ | $+5.2$ | $86.5 \pm 0.8$ | $+5.7$ | $90.4 \pm 0.3$ |

Table 4: **Best Accuracy after Sample Scaling beats Larger Model.** Acc indicates the highest accuracy achieved through scaling. Groups 1 & 2 are defined in Table 2. Gain represents the improvement over the best single-model accuracy reported in Table 2. For reference, we also include the performance of the large model Qwen-2.5 72B, showing that our composed small models can outperform it on GPQA and GSM8K.

## 5 DISCUSSION

**Mathematical Intuition behind SLM-MUX**. Different SLMs have complementary strengths: for any given question, some models are more likely to answer correctly than others. SLM-MUX exploits this by selecting the most self-consistent model's output through a simple rule-based mechanism that requires no inter-model communication.

The key insight is that the confidence score can identify the strongest model. We assume that for each question, there is a unique correct answer, while incorrect answers are scattered rather than clustered. Under this assumption, a model with higher accuracy $p_i$ produces the correct answer more frequently across $N$ samples, leading to a higher confidence score. Therefore, selecting the model with the highest confidence score effectively identifies the model most likely to be correct.

More formally, consider $K$ models where model $i$ has probability $p_i$ of being correct. Let $i^* = \arg\max_i p_i$ denote the strongest model with margin $\gamma = p_{i^*} - \max_{j \neq i^*} p_j > 0$. Under our assumption, the confidence score $s_i$ (the frequency of the most common answer over $N$ samples) concentrates around $p_i$. Applying Hoeffding's inequality and a union bound, the probability of correctly selecting the strongest model satisfies:

$$\Pr(\hat{i} = i^*) \geq 1 - 2(K-1)\exp\left(-\frac{N\gamma^2}{2}\right).$$

This bound shows that the probability of misidentifying the strongest model decays exponentially with sample size $N$.

This selection mechanism explains why SLM-MUX outperforms alternatives. Unlike a single fixed model, SLM-MUX performs per-question routing, effectively achieving accuracy $p_{\max}$ by always selecting the strongest available expert. Unlike pooling methods such as Agent Forest that aggregate outputs from all models, SLM-MUX avoids interference from weaker models. For instance, if the strongest model has $p_1 = 0.8$ and a weaker one has $p_2 = 0.3$, pooling their outputs merely dilutes the correct answer's frequency. By isolating the strongest model and selecting its most frequent answer, SLM-MUX preserves the full predictive power of the most reliable source. We provide a more detailed comparative analysis with self-consistency and Agent Forest in Appendix D.2.

**Extending SLM-MUX to Open-Ended Generation.** Although the current implementation of SLM-MUX relies on majority voting and is therefore restricted to tasks with discrete answer spaces, the underlying idea of selecting the most self-consistent model is more general. For open-ended generation, one can replace majority voting with alternative consistency estimators, such as LLM-as-a-judge scoring or embedding-based similarity measures. In Appendix E.1, we show a simple extension of SLM-MUX to HumanEval (Chen et al., 2021) using this idea and observe strong empirical gains.

**Extending SLM-MUX Beyond Generalist SLMs.** The experiments above focus on general-purpose SLMs. We further evaluate whether the consistency-based selection principle extends to other settings: (1) frontier LLMs such as GPT-4o and Gemini-2.5-Flash, and (2) domain-specific fine-tuned models such as code and math specialists. In both cases, SLM-MUX achieves consistent improvements over the best single model. Full experimental details are provided in Appendix E.

**Limitation and Future Work**. The SLM-MUX framework has two main limitations. First, its design is static and does not adapt to specific questions. For every query, it uses a fixed group of models that are pre-selected through exhaustive search – a method that is slow and costly when there are many models to choose from. When models are tied, the framework uses their past accuracy on a validation set to decide, which is also a fixed, non-adaptive rule. Second, the way the framework measures model confidence is simple. It relies only on self-consistency – how often a model produces the same answer. This can be a problem because a model can be very consistent while still being incorrect.

**Conclusion**. This work demonstrates that orchestration methods designed for frontier models paradoxically degrade the performance of SLMs by amplifying errors. To address this, we propose SLM-MUX, a framework that avoids inter-model discussion, instead selecting the most reliable output based on each model's self-consistency. We further introduce a model selection search algorithm to find complementary model combinations. Experiments show our method not only substantially outperforms existing strategies but also enables an ensemble of just two SLMs to surpass the much larger Qwen-2.5 72B model on key reasoning benchmarks. In summary, our work validates that intelligently orchestrating multiple efficient models—a "multi-core" approach—is a promising alternative to scaling monolithic models on the path toward more capable AI systems.

## ACKNOWLEDGMENTS

We thank Tom Griffiths from Princeton University, whose insights and feedback were instrumental in initiating this project.

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

## APPENDIX OVERVIEW

The appendix is organized into five main sections. **Section A** states the usage of LLMs in preparing this paper. **Section B** provides experimental details, including visual illustrations, single-model accuracies, and standard deviation calculations. **Section C** analyzes why discussion-based methods fail on SLMs, presenting groupthink analysis and prompt sensitivity studies. **Section D** validates the SLM-MUX design through consistency-accuracy correlation analysis, comparative analysis with voting-based methods, model selection search analysis, and test-set scaling validation. **Section E** demonstrates the generalization of SLM-MUX to open-ended generation, frontier LLMs, and domain-specific models. Finally, Section F provides dataset licenses.

## A  LLM USAGE STATEMENT

We used Cursor for coding. Large language models (LLMs) were employed to help polish drafts written by humans, and to assist in searching for related papers. The final choice of related work included in this paper was made entirely by the human authors after careful screening. LLMs were also used for proofreading and for providing suggestions.

## B  EXPERIMENTAL DETAILS

### B.1  VISUAL ILLUSTRATIONS OF SLM-MUX

To more effectively illustrate the workflow of our proposed composition method, we select several representative examples from the logs. We demonstrate them in Figure 10, Figure 11 and Figure 12.

**SLM-MUX surpasses majority voting in scenarios with initial disagreement among models.**. As illustrated by Figure 10, during the independent generation phase, Gemma-2-27B is the sole model to provide the correct answer. Hence, majority voting applied directly would fail to select the correct author.

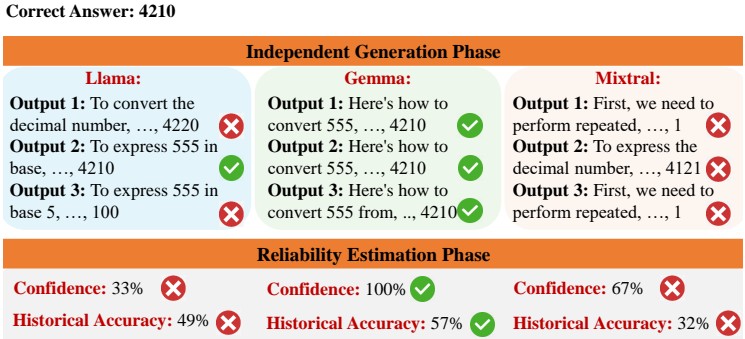

Figure 10: **An illustration of the SLM-MUX method applied to the MATH dataset.** In the independent generation phase, three models are used: LLaMA-3.1-8B (denoted as Llama), Gemma-2-27B (denoted as Gemma), and Mixtral-8×7B (denoted as Mixtral). Because the three models provide different answers at first, so each model is invoked two more times. Gemma obtains the highest confidence score and is therefore selected as the final output.

### B.2  ACCURACY OF SINGLE LLMS

We evaluated the accuracy of single model accuracy under the condition of temperature equal to zero. The results are shown in Table 5 and Table 6.

| Model | MATH Acc (%) | GPQA Acc (%) | GSM Acc (%) |
|---|---|---|---|
| Llama-3.1-8B | 48.6 | 23.7 | 84.2 |
| Mistral-8×7B | 31.6 | 31.9 | 63.4 |
| Gemma-2-27B | 56.8 | 38.8 | 81.6 |

Table 5: **Small Model Base Performance.** Base model accuracy on MATH, GPQA, and GSM8K.

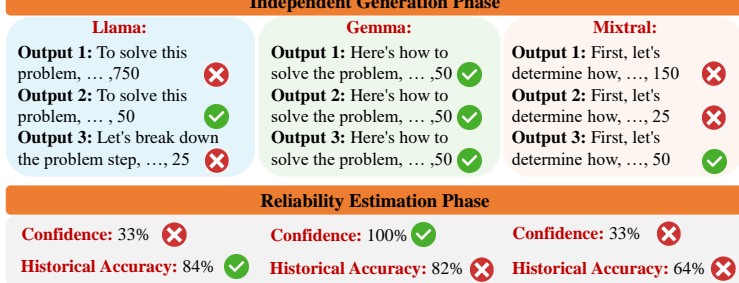

**Question:** Elvis has a monthly saving target of $1125. In April, he wants to save twice as much daily in the second half as he saves in the first half in order to hit his target. How much does he have to save for each day in the second half of the month?
**Correct Answer: 50**

| Independent Generation Phase | | |
|---|---|---|
| **Llama:** | **Gemma:** | **Mixtral:** |
| **Output 1:** To solve this problem, … ,750 ❌ | **Output 1:** Here's how to solve the problem, … ,50 ✅ | **Output 1:** First, let's determine how, …, 150 ❌ |
| **Output 2:** To solve this problem, … , 50 ✅ | **Output 2:** Here's how to solve the problem, … ,50 ✅ | **Output 2:** First, let's determine how, …, 25 ❌ |
| **Output 3:** Let's break down the problem step, …, 25 ❌ | **Output 3:** Here's how to solve the problem, … ,50 ✅ | **Output 3:** First, let's determine how, …, 50 ✅ |

| Reliability Estimation Phase | | |
|---|---|---|
| **Confidence:** 33% ❌ | **Confidence:** 100% ✅ | **Confidence:** 33% ❌ |
| **Historical Accuracy:** 84% ✅ | **Historical Accuracy:** 82% ❌ | **Historical Accuracy:** 64% ❌ |

Figure 11: **An illustration of the SLM-MUX method applied to the GSM8K dataset.** In the independent generation phase, different models produce different answers. However, when we invoke each model multiple times, we observe that Llama and Mixtral only yield correct answers approximately one-third of the time. In contrast, Gemma demonstrates stable performance.

**Question:** Question: A student regrets that he fell asleep during a lecture in electrochemistry, facing the following incomplete statement in a test: "Thermodynamically, oxygen is a …… oxidant in basic solutions. Kinetically, oxygen reacts …… in acidic solutions." Which combination of weaker/stronger and faster/slower is correct?
(A) weaker – slower
(B) stronger – slower
(C) weaker – faster
(D) stronger – faster
**Correct Answer: (A)**

| Independent Generation Phase | | |
|---|---|---|
| **Llama:** | **Gemma:** | **Mixtral:** |
| **Output 1:** Answer: C, Explanation: … ❌ | **Output 1:** Answer: D, … ❌ | **Output 1:** To answer this question, …, A ✅ |
| **Output 2:** Answer: A, In basic solutions, … ✅ | **Output 2:** Answer: D, … ❌ | **Output 2:** To answer this question, …, A ✅ |
| **Output 3:** Answer: D , In basic solutions, … ❌ | **Output 3:** Answer: D, … ❌ | **Output 3:** To answer this question, …, A ✅ |

| Reliability Estimation Phase | | |
|---|---|---|
| **Confidence:** 33% ❌ | **Confidence:** 100% ✅ | **Confidence:** 100% ✅ |
| **Historical Accuracy:** 24% ❌ | **Historical Accuracy:** 32% ❌ | **Historical Accuracy:** 39% ✅ |

Figure 12: **An illustration of the SLM-MUX method applied to the GPQA dataset.** During the independent generation phase, Gemma and Mixtral obtain the same confidence score. However, considering historical accuracy, Mixtral ranks higher. Therefore, Mixtral's answer is selected as the final output.

### B.3 STANDARD DEVIATION OF THE DATA POINTS IN FIGURE 5

Although all experiments are run in a deterministic setting with temperature set to zero, we can still compute the standard deviation of each datapoint by treating the outcome as a Bernoulli variable. Specifically, if there are $n_{\text{correct}}$ correct answers and $n_{\text{wrong}}$ incorrect answers, the standard deviation

| Model | MATH | | GPQA | |
|---|---|---|---|---|
| | Accuracy (%) | Token Usage | Accuracy (%) | Token Usage |
| DeepSeek V3 | 87.0 | 419,513 | 55.1 | 173,885 |
| Gemini 2.0 Flash | 90.4 | 361,737 | 63.6 | 195,576 |
| GPT-4o | 79.8 | 408,410 | 51.0 | 212,037 |

Table 6: **Large Model Base Performance.** Base model performance and token usage on MATH and GPQA datasets. Accuracy is the percentage of correct answers, and token usage reflects total tokens consumed (prompt + response) over the entire dataset for each model.

is

$$\frac{\sqrt{\text{Var}(X)}}{\sqrt{n_{\text{total}}}} = \frac{\sqrt{p(1-p)}}{\sqrt{n_{\text{total}}}} = \frac{\sqrt{\frac{n_{\text{correct}}}{n_{\text{total}}}\left(1 - \frac{n_{\text{correct}}}{n_{\text{total}}}\right)}}{\sqrt{n_{\text{total}}}},$$

where $n_{\text{total}} = n_{\text{correct}} + n_{\text{wrong}}$.

The results are summarized in Table 7.

Table 7: Accuracy and estimated standard deviation on MATH ($n = 500$) and GPQA ($n = 196$) using datapoints from Figure 5.

| | MATH ($n = 500$) | | GPQA ($n = 196$) | |
|---|---|---|---|---|
| Method | SLM orchestration | LLM composition | SLM orchestration | LLM composition |
| Single Model Max | $56.8 \pm 2.22$ | $90.4 \pm 1.32$ | $38.8 \pm 3.48$ | $63.6 \pm 3.44$ |
| Mixture-of-Agents | $46.2 \pm 2.23$ | $88.8 \pm 1.41$ | $33.3 \pm 3.37$ | $58.6 \pm 3.52$ |
| LLM-Debate | $51.6 \pm 2.23$ | $90.8 \pm 1.29$ | $36.9 \pm 3.45$ | $65.6 \pm 3.39$ |
| Multi-Agent Verification | $48.4 \pm 2.23$ | $91.6 \pm 1.24$ | $35.4 \pm 3.42$ | $64.2 \pm 3.42$ |

## C  WHY DISCUSSION-BASED METHODS FAIL ON SLMS

### C.1  GROUPTHINK ANALYSIS

We analyze the experiment logs of LLM-Debate using small language models (SLMs) in Section 4.1. Among 500 debate problems, 242 resulted in failure (48.4%). For each of the 242 failed debates, we first used an analyzer LLM to produce a process-focused failure analysis. We then used a separate labeling LLM to classify whether each failed debate was due to groupthink.

The labeling results are shown in Table 8:

These results reinforce our claim that groupthink is a major failure mode in SLM-based LLM-debate.

We provide the exact prompts used by (i) the analyzer LLM to generate the 242 failure analyses (Figure 13) and (ii) the groupthink labeler LLM to classify groupthink (Figure 14). Placeholders such as `{problem}` indicate runtime substitutions by our code.

### C.2  PROMPT SENSITIVITY ANALYSIS

A natural concern is whether the performance gap between discussion-based methods and SLM-MUX is due to suboptimal prompt selection rather than inherent limitations. To address this, we conduct a prompt sensitivity analysis on the Mixture-of-Agents (MoA) baseline using the MATH benchmark.

We use Gemini-2.5-Flash to generate 10 diverse aggregator prompts for MoA. Table 9 summarizes the results. Our baseline prompt (46.2%) outperforms the average of the tuned prompts (41.8%) and surpasses 7 out of 10 generated prompts. Even the best tuned prompt (48.4%) remains substantially below the best single model (56.8%) and SLM-MUX (61.8%).

To further stress-test this result, we conducted iterative prompt optimization directly on the test set for 6 rounds, selecting the best-performing prompt at each iteration. Even under this extremely favorable setting for MoA, the accuracy peaked at 55.6%, still below the best single model (56.8%) and far below SLM-MUX (61.8%). This confirms that the performance gap reflects inherent limitations of discussion-based aggregation when applied to SLMs, rather than an artifact of prompt selection.

## D  VALIDATION OF SLM-MUX DESIGN

### D.1  CONSISTENCY VS ACCURACY CORRELATION

We empirically study how per-question self-consistency correlates with accuracy on four datasets: GSM8K, MATH, GPQA, and HUMANEVAL. For each model–dataset pair, we compute a self-consistency score for every question (as defined in the main text) and group questions into three bins according to this score: *Low* $[0.0, 0.5)$, *Medium* $[0.5, 0.8)$, and *High* $[0.8, 1.0]$. We then measure the empirical accuracy (fraction of correct answers) within each bin.

```
As an expert in analyzing multi-agent AI systems, your task is to
    analyze why an 'LLM Debate' process failed to find the correct
     answer. Your focus should be on the *debate dynamics and
    process*, not just the mathematical details. The goal is to
    understand the failure of the debate methodology itself.

**Ground Truth:**
- **Problem Statement:** {problem}
- **Correct Answer:** {ref_answer}

**Debate Information:**
- **Final Incorrect Answer from System:** {system_answer}

**Analysis of Round 1:**
- **Model `{model_name}` proposed:**
  - Answer: `{extracted_answer}`
  - Reasoning:
```
{full_text}
```

... (repeats per round and per model)

**Your Analysis Task:**
Based on the debate history, provide a "Debate Failure Analysis".
   Do not focus on simple calculation mistakes. Instead, analyze
   the interaction between the models and the structure of the
   debate. Pinpoint the core reasons the *debate process* failed.
    Consider these questions:

1.  **Error Propagation vs. Correction:** How did initial errors
    influence later rounds? Were there moments where a correct
    idea was introduced but ignored or overruled? Why did the
    debate fail to self-correct?
2.  **Groupthink and Influence Dynamics:** Did the models converge
     on a flawed consensus? Did one or more influential but
    incorrect models lead the group astray? Was there evidence of
    independent reasoning that was shut down?
3.  **Argumentation Quality:** Did the models provide convincing
    but ultimately flawed arguments? Did they effectively
    challenge each other's reasoning, or was the debate
    superficial?
4.  **Critical Failure Point in the Debate:** Identify the single
    most critical turn or moment in the debate that sealed its
    failure. What happened, and why was it so impactful?
5.  **Improving the Debate:** What is the single most important
    change to the debate protocol or dynamics that could have
    prevented this failure? (e.g., different communication rules,
    promoting dissident opinions, etc.)

Provide a concise, expert analysis focusing on the *process*
    failure.
```

Figure 13: Prompt Template for Failure Analysis.

| Metric | Count | Rate |
|---|---|---|
| Total Debates Analyzed | 500 | 100% of total |
| Failed Debates (System Error) | 242 | 48.4% of total |
| *Breakdown of Failed Debates:* | | |
|    Attributed to Groupthink | 144 | 59.5% of failures |
|    Attributed to Other Causes | 79 | 32.6% of failures |
|    Classification Unsuccessful | 19 | 7.9% of failures |

Table 8: **Failure Cause Attribution** This table shows the cause attribution for LLM-Debate when involving SLMs.

```
You are an expert analyst of multi-agent LLM debates. Your goal is
   to determine whether the failure primarily involved
   groupthink/conformity dynamics. Groupthink indicators include:
    early flawed consensus, explicit capitulation to a majority,
   social proofing, adopting peers' answers without critique,
   abandoning independent reasoning to match others, or
   reinforcing an incorrect majority despite available dissent.
   Not-groupthink includes failures due to independent arithmetic
   /logic errors, argument complexity/veneer effects without
   convergence, or chaotic divergence with no consensus influence
   . Return STRICT JSON only, with keys: groupthink (bool),
   confidence (float 0-1), reasons (string), cues (array of
   strings).
```

Figure 14: Prompt for Groupthink Classification.

Figure 15 reports the resulting accuracies for two representative SLMs on each dataset. Across GSM8K, MATH, and HUMANEVAL we observe a strong positive relationship between self-consistency and accuracy: questions in the high-consistency bin are substantially more likely to be answered correctly than those in the low-consistency bin. GPQA exhibits a weaker but still positive correlation. Overall, these results provide empirical support for the link between self-consistency and correctness assumed in our method.

### D.2   COMPARATIVE ANALYSIS WITH VOTING-BASED METHODS

Since SLM-MUX also involves voting on model outputs, we examine its differences from standard self-consistency and Agent Forest to better explain the source of our improvements.

To explain our stronger performance, we note a limitation of self-consistency methods. Suppose a model has probability $p$ of answering a question correctly. When self-consistency samples $N$ responses, the probability of obtaining the correct answer after majority voting follows a binomial distribution:

$$A(N, p) = \Pr\left(X \geq \left\lceil \tfrac{N}{2} \right\rceil\right) = \sum_{k=\lceil N/2 \rceil}^{N} \binom{N}{k} p^k (1-p)^{N-k}, \quad X \sim \text{Binomial}(N, p) \qquad (1)$$

Table 9: Prompt sensitivity analysis for MoA on MATH. The baseline prompt used in our experiments is already near-optimal.

| Setting | Accuracy |
|---|---|
| SLM-MUX (Ours) | **61.8%** |
| Best Single Model | 56.8% |
| Tuned Prompt (Best) | 48.4% |
| Baseline Prompt (Used in Paper) | 46.2% |
| Tuned Prompt (Average) | 41.8% |
| Tuned Prompt (Worst) | 25.8% |

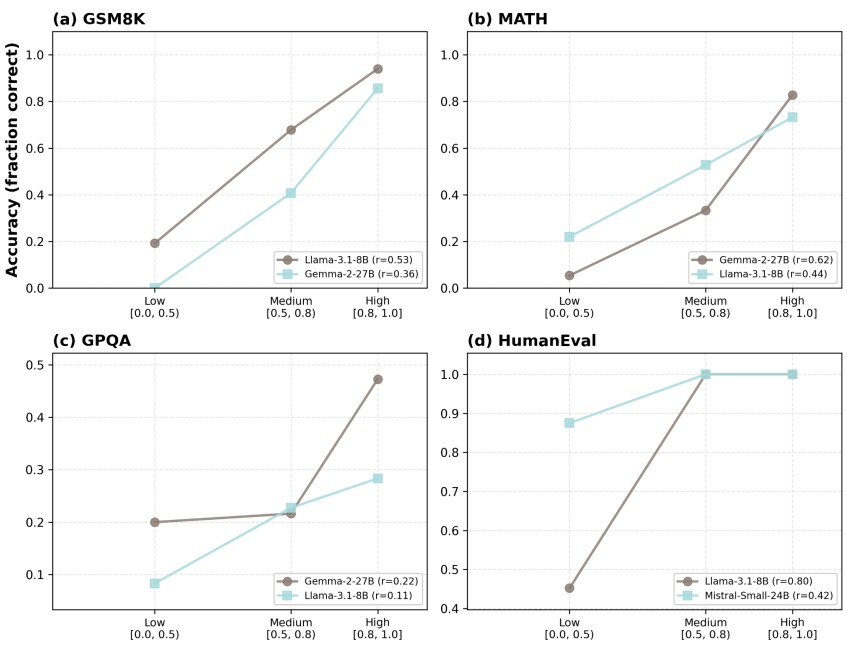

Figure 15: Accuracy as a function of per-question self-consistency score on GSM8K, MATH, GPQA, and HUMANEVAL. Questions are grouped into three bins by self-consistency: *Low* $[0.0, 0.5)$, *Medium* $[0.5, 0.8)$, and *High* $[0.8, 1.0]$. Each line corresponds to a different SLM; legends report the Pearson correlation $r$ between self-consistency and correctness.

We observe that $A(N, p)$ exceeds $p$ only when $p > 0.5$, meaning self-consistency is effective only in this regime. When $p < 0.5$, self-consistency can actually lower overall accuracy.

For any dataset, we can conceptually divide examples into three types of questions. **Type 1**: $p = 100\%$, the model always answers correctly. **Type 2**: $p > 50\%$, the model is more likely than not to be correct. **Type 3**: $p < 50\%$, the model is more likely to be wrong. The overall effect of self-consistency is then the improvement from Type 2, offset by the degradation from Type 3. Improvement occurs only when the dataset contains a sufficiently large proportion of Type 2 questions.

For SLM-MUX, we select the output from the most confident model, so the accuracy can be approximated as $A(N, p_{\max})$, where $p_{\max}$ is the highest probability among the participating models. By routing to the model with the highest $p_{\max}$ on each question, we effectively enlarge the proportion of Type 2 questions, leading to higher overall accuracy.

For the Agent Forest approach, answers are drawn evenly from all models, so its accuracy can be approximated as $A(N, \bar{p})$, where $\bar{p}$ is the average probability across models. This generally results in lower accuracy than SLM-MUX, as weaker models dilute the signal from stronger ones.

## D.3 MODEL SELECTION SEARCH ANALYSIS

### D.3.1 SEARCH-SET SIZE STABILITY

We analyze how the size of the search set (number of problems used in the search phase) affects the resulting model ranking. For each of the three benchmarks MATH, GSM8K, and GPQA, we treat the full benchmark (approximately 500 problems) as the search pool and first run our search procedure on the full set to obtain a ranking of all candidate models. We then record the models occupying Rank 1, Rank 2, and Rank 3 under this full-set ranking.

Next, we subsample the search set to smaller sizes and re-run the search. Specifically, for each dataset we evaluate search-set sizes of 100, 200, 300, 400, and the full set. At each size, we recompute the ranking over all models and track the ranks of the three models that were Rank 1–3 under the full-set setting.

Figure 16 shows the results. Each curve corresponds to one of the Rank 1/2/3 models under the full-set ranking, and the $y$-axis reports its rank when the search-set size is changed. Across all three

datasets, these models consistently remain within the top three positions even when the search set is reduced to as few as 100 problems, with only minor swaps in their relative order on MATH and GSM8K. This suggests that a search set of a few hundred problems is sufficient to stably identify the top-performing models.

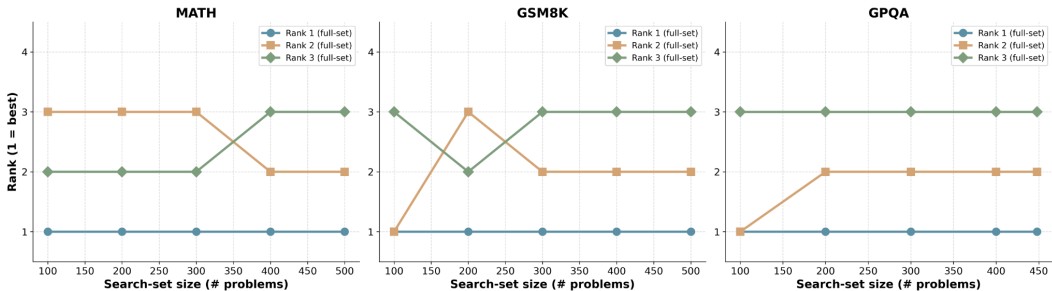

Figure 16: Stability of model rankings with respect to search-set size on MATH, GSM8K, and GPQA. For each dataset, we first determine the top three models using the full search set and then track their ranks as the search-set size is reduced.

### D.3.2   RANDOM MODEL SELECTION BASELINE

A natural question is whether the performance gains of SLM-MUX stem from the model selection search or from the orchestration architecture itself. To isolate the contribution of the architecture, we conduct an experiment where model combinations are selected randomly rather than through our search procedure.

For each dataset, we randomly sample model combinations of size $K = 2, 3, 4$ from our pool of five SLMs and apply SLM-MUX. We compare the resulting accuracy against the best single model within each random pool. Table 10 reports the average performance across all random combinations.

Table 10: Performance of SLM-MUX with randomly selected model combinations. Even without optimized model selection, SLM-MUX consistently outperforms the best single model in the pool.

| Dataset | $K$ | SLM-MUX (Random) | Best Single Model | $\Delta$ (Gain) |
|---|---|---|---|---|
| MATH-500 | 2 | 67.2% | 66.1% | +1.1 |
| | 3 | 71.5% | 69.8% | +1.7 |
| | 4 | 73.7% | 71.8% | +1.9 |
| GSM8K | 2 | 84.1% | 81.7% | +2.4 |
| | 3 | 87.3% | 83.6% | +3.7 |
| | 4 | 88.8% | 84.0% | +4.8 |
| GPQA | 2 | 43.0% | 41.3% | +1.7 |
| | 3 | 44.5% | 44.4% | +0.1 |
| | 4 | 46.3% | 46.0% | +0.3 |

As shown in Table 10, SLM-MUX consistently outperforms the best single model even when the model combination is selected randomly. On MATH-500 and GSM8K, the gains are substantial (up to +4.8 on GSM8K with $K = 4$). On GPQA, where consistency is a weaker signal for correctness, the gains are smaller but still positive.

To further quantify how often SLM-MUX improves over single models, we compute the **Effective Combination Rate**: the percentage of all possible $K$-model subsets where SLM-MUX outperforms the best single model in that subset. Table 11 reports the results.

On GSM8K, 100% of combinations are effective across all values of $K$. On MATH-500 and GPQA, the effective rate increases with $K$, reaching 100% at $K = 4$. Even at $K = 2$, the majority of combinations (60–100%) are effective. These results demonstrate that the space of "workable" model combinations is dense, and one does not need to search extensively to find an effective subset. The

Table 11: Effective Combination Rate: percentage of model combinations where SLM-MUX outperforms the best single model in the subset.

| Dataset | $K = 2$ | $K = 3$ | $K = 4$ |
|---------|---------|---------|---------|
| MATH-500 | 70% | 100% | 100% |
| GSM8K | 100% | 100% | 100% |
| GPQA | 60% | 80% | 100% |

model selection search provides additional gains by identifying the optimal combination, but the architecture is robust even without it.

### D.4 TEST-SET VALIDATION OF SCALING BEHAVIOR

In Section 3.3, we evaluated the "Adding More Participating Model Types" scaling dimension on the validation set. Here we report the corresponding results on the test set to verify that the observed trends generalize.

Table 12 summarizes the test-set accuracy as the number of participating models $K$ increases from 1 to 5. For each value of $K$, we use the model combination selected by the search procedure described in Section 3.2. Figure 17 visualizes these results.

Table 12: Test-set accuracy (%) as the number of participating models $K$ increases. For $K = 1$, the best single model is reported.

| $K$ | GPQA | GSM8K | MATH |
|-----|------|-------|------|
| 1 (best single) | 47.98 | 90.20 | 78.20 |
| 2 | 50.21 | 91.51 | 80.27 |
| 3 | 51.81 | 91.29 | 81.09 |
| 4 | 49.16 | 91.44 | 81.00 |
| 5 | 45.18 | 90.80 | 80.64 |

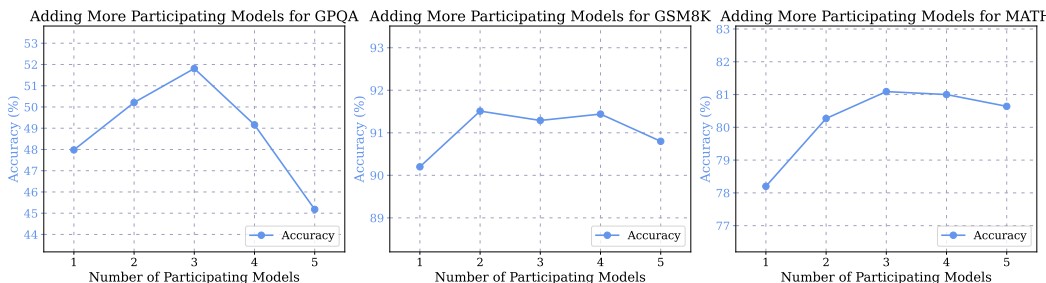

Figure 17: Test-set accuracy as a function of the number of participating models on GPQA, GSM8K, and MATH. The trends mirror those observed on the validation set: GPQA peaks at $K = 3$, GSM8K saturates quickly, and MATH shows continued improvement up to $K = 3$.

## E GENERALIZATION OF SLM-MUX

### E.1 OPEN-ENDED GENERATION (HUMANEVAL)

In the main text, SLM-MUX is instantiated on tasks with discrete answer spaces, where self-consistency can be measured via majority voting over sampled outputs. To check whether the same principle extends to open-ended generation, we apply SLM-MUX to the HUMANEVAL code-generation benchmark.

**Consistency estimator for open-ended generation.**. On HUMANEVAL, exact-string majority voting is not appropriate, so we replace it with a semantic consistency estimator. For each model and each problem, we sample $N = 5$ code generations with temperature 0.3. We then encode the 5 generations using the pretrained embedding model `Salesforce/codet5p-110m-embedding` and compute pairwise cosine similarities, yielding a $5 \times 5$ similarity matrix. From this matrix,

Table 13: Pass@1 on HUMANEVAL for individual SLMs.

| Model | Pass@1 |
|---|---|
| Llama-3.1-8B-Instruct | 0.178 |
| Qwen2.5-7B-Instruct | 0.485 |
| Mistral-Small-24B | 0.870 |
| Qwen2.5-Coder-7B | 0.893 |

Table 14: SLM-MUX on HUMANEVAL (Pass@1). Each row corresponds to a pair of SLMs; SLM-MUX selects the output from the model with higher embedding-based consistency.

| Setup | Models combined | SLM-MUX (Pass@1) |
|---|---|---|
| Exp 1 | Llama-3.1-8B-Instruct + Qwen2.5-7B-Instruct | **0.506** |
| Exp 2 | Mistral-Small-24B + Qwen2.5-Coder-7B | **0.939** |

we identify the most coherent cluster of generations (of size $x$) and use $x/5$ as the model's self-consistency score for that problem, analogous to the confidence score derived from majority voting in the discrete-answer setting.

We also experimented with an LLM-as-a-judge–based consistency estimator (using Qwen2.5-7B-Instruct as the judge) and found that the embedding-based estimator exhibits a stronger correlation with ground-truth correctness (Pass@1). All results below therefore use the embedding-based consistency score.

**Results.**. Table 13 reports the Pass@1 of each individual SLM on HUMANEVAL. Table 14 reports the Pass@1 of SLM-MUX when combining two models at a time; for each problem, SLM-MUX selects the solution from the model with the larger embedding-based consistency score.

### E.2 FRONTIER LLMS

We evaluate whether SLM-MUX can exploit complementary strengths between state-of-the-art frontier models. We pair GPT-4o with Gemini-2.5-Flash and apply SLM-MUX on MATH-500, GPQA, and GSM8K-500. For each problem, we sample $N = 5$ responses per model at temperature 0.3 and apply self-consistency routing: for each problem, we perform majority voting within each model's samples, then route to the model showing higher agreement.

Table 15 summarizes the results. As a reference, "Perfect Routing" indicates the theoretical upper bound achievable if the system always selects the correct model when at least one succeeds.

Table 15: SLM-MUX performance when applied to frontier LLMs (GPT-4o and Gemini-2.5-Flash).

| Benchmark | GPT-4o | Gemini-2.5-Flash | SLM-MUX | Perfect Routing |
|---|---|---|---|---|
| MATH-500 | 73.0% | 92.1% | 92.8% | 94.2% |
| GPQA | 50.7% | 51.1% | **60.1%** | 73.7% |
| GSM8K-500 | 89.1% | 85.7% | 89.4% | 91.4% |

The results reveal two distinct regimes. On GPQA, the two models exhibit complementary error patterns, and SLM-MUX achieves 60.1% accuracy, surpassing the best single model by nearly 10 percentage points. This demonstrates that SLM-MUX effectively exploits complementary strengths even at the frontier scale. On MATH and GSM8K, the Perfect Routing bounds (94.2% and 91.4%) are only marginally higher than the single-model baselines, indicating high overlap in the models' correct predictions. The limited gains on these benchmarks reflect this ceiling rather than a limitation of the routing mechanism.

### E.3 DOMAIN-SPECIFIC FINE-TUNED MODELS

Domain-specific fine-tuned models are widely deployed in practice. We evaluate whether SLM-MUX can effectively orchestrate such specialized models by testing on two domains: code generation and mathematical reasoning.

Table 16: SLM-MUX performance when orchestrating domain-specific fine-tuned models.

| Domain | Benchmark | Best Single Model | SLM-MUX | $\Delta$ (Gain) |
|---|---|---|---|---|
| Code Generation | HumanEval | 89.3% | **93.9%** | +4.6 |
| Math Reasoning | MATH-500 | 58.8% | **62.2%** | +3.4 |

For code generation, we pair Qwen2.5-Coder-7B (a code-specialized model) with Mistral-Small-24B (a general-purpose model) on HumanEval. We use the same embedding-based consistency estimator described in Section E.1. For mathematical reasoning, we pair DeepSeek-Math-7B-RL (a math-specialized model) with Llama-3.1-8B-Instruct on MATH-500, using standard majority voting for consistency estimation. Both experiments sample $N = 5$ responses per model at temperature 0.3.

As shown in Table 16, SLM-MUX achieves consistent improvements in both domains. On HumanEval, the orchestrated system reaches 93.9% Pass@1, outperforming the code specialist (89.3%) by 4.6 percentage points. On MATH-500, combining the math specialist with a general-purpose model yields 62.2% accuracy, a 3.4pp improvement. These results demonstrate that SLM-MUX generalizes effectively to domain-specific fine-tuned models, successfully capturing complementary strengths between specialists and generalists.

## F    LICENSES FOR DATASETS

The MATH dataset is licensed under the MIT License.
The GPQA dataset is licensed under the Creative Commons Attribution 4.0 International (CC BY 4.0) License.
The GSM8K dataset is licensed under the MIT License.

