# OpenReview forum: "SLM-MUX: Orchestrating Small Language Models for Reasoning"
_ICLR.cc/2026/Conference — ICLR 2026 Poster_

### Official Review · Reviewer_7wuE · 2025-10-28

**Soundness:** 3
**Presentation:** 3
**Contribution:** 2
**Rating:** 4
**Confidence:** 3

**Summary:**

This paper proposes an orchestration framework for small language models (SLMs), called SLM-MUX, and further explores test-time scaling strategies specifically for this method. The authors show that SLM-MUX can effectively coordinate multiple SLMs and achieves strong performance across various mathematical reasoning tasks.

Their main contributions are as follows:
1. They identify the limitations of applying traditional orchestration methods, originally developed for large language models (LLMs), to SLMs.
2. They introduce SLM-MUX, an orchestration approach that works particularly well for SLMs compared to discussion-based methods.
3. They investigate strategies for model selection and search–compute scaling, tailored to the SLM-MUX framework.

**Strengths:**

The authors identify an interesting problem: existing orchestration methods developed for large language models (LLMs), such as discussion-based approaches, do not directly transfer well to small language models (SLMs).

The paper provides extensive empirical evaluations and complementary mathematical analyses to study the proposed SLM-MUX framework.

**Weaknesses:**

The proposed method, including its model selection objectives, appears somewhat heuristic. While the consistency of sampled answers may indeed correlate with correctness, relying on it as the primary criterion for selecting the best responses seems somewhat ad hoc. A stronger theoretical justification would help strengthen confidence in this approach. The analysis presented in the discussion section is a good starting point, but it would benefit from being expanded and made more rigorous to support the claims more convincingly.

Regarding the contradiction penalty, it also comes across as rather ad hoc, as it penalizes models that consistently produce incorrect answers. Overall, these components feel more like empirical adjustments designed to boost performance metrics rather than principled elements of the framework.

**Questions:**

I understand that this method requires a validation set for model selection and related adjustments. Do the baseline methods (such as the discussion-based approaches) also rely on a validation set? How do you ensure that the demonstrated advantage is not due to access to additional information, or better overfitting capability?

---

> ### Author Response · Authors · 2025-11-27
>
> We thank the reviewer for their constructive and helpful review. We respond to concerns below and have also updated the paper (revisions highlighted).
>
> > The proposed method, including its model selection objectives, appears somewhat heuristic. While the consistency of sampled answers may indeed correlate with correctness, relying on it as the primary criterion for selecting the best responses seems somewhat ad hoc. A stronger theoretical justification would help strengthen confidence in this approach. The analysis presented in the discussion section is a good starting point, but it would benefit from being expanded and made more rigorous to support the claims more convincingly.”
>
> Thanks for point out, here we want to provide two updates to address this:
>
> 1. Updated Theoretical Analysis:
>
> We have expanded the Discussion section to provide a better proof of our method's validity. We prove that given $K$ LMs, the confidence score is likely to be highest for the model with high probability to generate correct answer.
>
> In the revised paper, we formally establish that as the number of consistency samples $K_{sc}$ increases, the probability of selecting the best model (the one with accuracy $p_{max}$) converges to 1 exponentially fast:
>
> $$\mathbb{P}(\text{Selection} = \text{Best Model}) \ge 1 - 4K \exp\left(-\frac{K_{sc} \gamma^2}{8}\right)$$
>
> where $\gamma$ is the performance margin between the best and second-best models. This confirms that our approach consistently identifies correct answers even without access to test set labels.
>
> 2. Empirical Verification
> We systematically verified the correlation between Consistency and Correctness. On discrete tasks like MATH and GSM8K, we observe strong positive correlations (Pearson $r \in [0.36, 0.62]$). Our primary focus is not limited to discrete settings, but rather validating the robustness of the principle across domains. We extended this verification to HumanEval. Even in this open-ended domain, using embedding-based similarity as a consistency metric yields a clear correlation with code correctness. This confirms that "consistency implies correctness" is a robust principle rather than an ad hoc observation. The results of this ablation is shown in Figure 15 in the updated paer.

---

> ### Author Response · Authors · 2025-11-27
>
> > “Regarding the contradiction penalty, it also comes across as rather ad hoc, as it penalizes models that consistently produce incorrect answers. Overall, these components feel more like empirical adjustments designed to boost performance metrics rather than principled elements of the framework.”
>
> We would like to clarify that the contradiction penalty is not merely an empirical adjustment, but a principled component designed to establish a Reliability Lower Bound.
> We have revised Section 3.2 in the main text to explicitly articulate this design philosophy:
> Maximize Accuracy ($\lambda \to 0$): Pursues the Optimistic Upper Bound of the ensemble's potential.
> Minimize Contradiction ($\lambda \to 1$): Enforces a Pessimistic Lower Bound by strictly penalizing confident errors.
> Our objective function is formally constructed to balance these two theoretical bounds. The contradiction penalty serves as a necessary regularization term to prevent the selection of models that are "confidently wrong,". This dual-bound framework provides a robust theoretical justification for the penalty, moving beyond simple performance tuning.

---

> ### Author Response · Authors · 2025-11-27
>
> > “I understand that this method requires a validation set for model selection and related adjustments. Do the baseline methods (such as the discussion-based approaches) also rely on a validation set? How do you ensure that the demonstrated advantage is not due to access to additional information, or better overfitting capability?”
>
> Yes we use a validation set, but we point out that the performance advantage of SLM-MUX is structural and not a result of unfair optimization settings. As shown in Table 1 in the paper, SLM-MUX achieves good performance even before applying model selection.
>
> Additionally, in order to provide stronger evidence, we conducted two ablations to see if discussion-based methods (on SLMs) can outperform SLM-MUX when given access to a training set. In Figure 5, we reported the comparison of MoA and SLM-MUX, which we represent in the data below:
>
> | Method | Accuracy | Comparison |
> | :--- | :---: | :--- |
> | **Best Single Model** | **56.8%** | Standard Baseline |
> | MoA (Mixture-of-Agents) | 46.2% | **Degrades performance (-10.6%)** |
> | **SLM-MUX (Ours)** | **61.8%** | **Improves performance (+5.0%)** |
>
> Here, starting from this MoA datapoint, we conducted two prompt tuning experiments directly on the test set, which is a modified setting that is biased towards helping MoA achieve higher performance.
>
> First, we tested zero-shot prompt optimization. We generated 10 diverse aggregator prompts using Gemini-2.5-Flash for MoA on MATH. Among the 10 prompts, that the best variant only reached 48.4% (vs. baseline 46.2%), with an average of 41.8% and min accuracy of 25.8%.
>
> Second, we conducted an aggressive iterative prompt tuning test where we use evolution search to optimized the aggregator prompt directly on the test set. We generated 6 candidate prompts in within each iteration and repeated 6 iterations. The accuracy peaked at 56.8%, still significantly lower than SLM-MUX.
>
> Both of these experiments show that the advantage of SLM-MUX does not simply stem from access to validation dataset.

---

### Official Review · Reviewer_2yhV · 2025-10-31

**Soundness:** 2
**Presentation:** 2
**Contribution:** 2
**Rating:** 4
**Confidence:** 3

**Summary:**

The authors propose an SLM orchestration method called SLM-MUX. The authors show that orchestration methods that work for LLMs interestingly do not work when orchestrating SLMs. The authors propose a simple orchestration framework for SLMs based on picking the answers which are the most self-consistent after multiple generations. The authors provide an SLM model selection protocol for identifying sets of models which are able to orchestrate well. In mathematical reasoning datasets the authors demonstrate their approach outperforms existing orchestration methods.

**Strengths:**

* The paper is well written and easy to follow.

* The authors provide a detailed empirical analysis to demonstrate the effectiveness of SLM-MUX versus baselines. Also they additionally provide ablations of important components such as the trade-offs in the model selection objectives and the performance against the number of predictions made by the SLMs.

**Weaknesses:**

* The contribution of the paper is limited. SLM-MUX is doing a maximum consistency selection over SLMs which is open to abuse from SLMs that are consistently wrong on a test set (see question below)? Additionally SLM-MUX requires a model search prior to being used to remove SLMs which are highly confident in their wrong answers, so do we require a training set?
* The mathematical analysis needs development: I think I understand that using $p_{max}$ versus $\bar{p}$ like in AgentForest results in a higher number of correct answers? I think this deserves a bit more development. Like how is it related to multiple models? You mention $p_{max}$ is the max of three models (in line 466) which is rather ad-hoc, this could be any number of models. However you cannot evaluate $p$ without a training set? If you only have a test set you will consistently pick wrong answers potentially and you cannot assess $p_{max}$?
* The results in Figure 7 are rather disappointing showing that 2 SLMs maximize the score which balances union accuracy while minimizing the contradiction penalty. I feel that this is a fundamental limitation of this LM orchestration technique that if you do not collaborate to come up with an answer, there will inevitably be some contradictions. So I can see that having 2 SLMs will reduce this contradiction in the responses, however this also feels sub-optimal. We should be in a state that adding more SLMs strictly improves performance.
* Inconsistencies in the evaluations. Some results are reported over multiple runs with means and standard deviations e.g. Table 1. While others which are important such as Figure 5 which demonstrate how existing orchestration methods work for LLMs but not for SLMs only report a single seed. For instance it looks like the existing orchestration methods don’t work either for LLMs, but with no error bars to assess the noise in the experimental process this is difficult to conclude.

**Questions:**

* I’m confused by the definition of the Contradiction metric. How does it generalize to more than two models. Surely encouraging contradiction enables a more diverse set S for SLM-MUX, in lines 322-323 you specifically mention that diverse outputs enables SLM-MUX’s abilities. Did you try varying the model selection objective? A good comparison for the model selection criterion would be simply a random selection of SLMs.

* What would happen if you have no training set and you want to evaluate SLM-MUX 0-shot on a new test set? In algorithm 1 you do not have any validation accuracies or ground truth labels $y$ to select the most consistent model?

* What does SC signify in the row Single-Best-SC in Table 1?

---

> ### Author Response · Authors · 2025-11-27
>
> We thank the reviewer for their constructive and helpful review. We respond to concerns below and have also updated the paper (revisions highlighted).
>
> > “The contribution of the paper is limited. SLM-MUX is doing a maximum consistency selection over SLMs which is open to abuse from SLMs that are consistently wrong on a test set (see question below)? Additionally SLM-MUX requires a model search prior to being used to remove SLMs which are highly confident in their wrong answers, so do we require a training set?”
>
> For the first point, "open to abuse from SLMs that are consistently wrong on a test set (see question below)", we would like to give the following clarifications
>
> **1. Correlation between Confidence and Accuracy**
> While we agree that there exists theoretical risks of models being consistently wrong, but we want to point out that in practice, confidence score (consistency) serves as a reasonable proxy for correctness. To validate this, we conducted an ablation study across four diverse benchmarks: MATH, GSM8K, GPQA, and HumanEval. As shown in the **Figure 15 in the updated pdf", we observe a positive correlation between the model confidence score and accuracy across all examined datasets. This empirical evidence supports the validity of maximum consistency selection in our setting.
>
>
> **2. Role of Model Selection**
> We agree that on some benchmarks such as GPQA where consistency is weakly correlated to accuracy. Directly applying SLM-MUX without model selection can lead to suboptimal performance. As we show in Table 1 in our paper, SLM-MUX achieves marginal improvement on GPQA. However, we want to point out that model selection search is very important in overcoming this. By penalizing overconfident SLMs, it helps identify model combinations that leads to high accuracy.
>
> Towards the second point, "Additionally SLM-MUX requires a model search prior to being used to remove SLMs which are highly confident in their wrong answers, so do we require a training set?"
>
> Yes, SLM-MUX needs a validation set to conduct model selection search, but we point out that the process is data-efficient and robust to the dataset size. To demonstrate this, we conducted a sensitivity analysis by reducing the search size from 500 to 100 examples. As shown in figure below, the ranking of the top model combinations remains highly stable. The best combinations identified with 500 examples consistently appear at the top even when the search size is reduced to 100. This indicates that SLM-MUX can identify optimal combinations with limited data. Moreover, even when we allocate the same training set to baselines for prompt tuning, SLM-MUX consistently outperforms them. This result is shown in **Figure 16** in the updated pdf.

---

> ### Author Response · Authors · 2025-11-27
>
> > “The mathematical analysis needs development: I think I understand that using  versus like in AgentForest results in a higher number of correct answers? I think this deserves a bit more development. Like how is it related to multiple models? You mention  is the max of three models (in line 466) which is rather ad-hoc, this could be any number of models. However you cannot evaluate  without a training set? If you only have a test set you will consistently pick wrong answers potentially and you cannot assess ?”
>
>
> We thank the reviewer for encouraging a deeper mathematical analysis. We acknowledge that our initial example focused on a simplified case ($N=3$), we have expanded the theoretical proofs in the revised paper to cover the general setting. We now prove that for any number of models $K$, our method systematically outperforms the baseline. Specifically, we address the concerns in three steps:
>
> 1. **Generalization to K Models:** We have removed the assumption of three models and derived general bounds for an ensemble of size $K$.
>
> 2. **Convergence without Test Labels:** We acknowledge the concern regarding evaluation without ground truth. SLM-MUX does not require ground truth labels at test time to identify $p_{max}(S)$. By utilizing Hoeffding's inequality, we prove that the empirical confidence score (measured via self-consistency) serves as a reliable proxy for true accuracy.
>
> 3. **Theoretical Guarantee:** In the revised paper, we formally establish that as the number of consistency samples $K_{sc}$ increases, the probability of selecting the best model (the one with accuracy $p_{max}$) converges to 1 exponentially fast:
>
> $$\mathbb{P}(\text{Selection} = \text{Best Model}) \ge 1 - 4K \exp\left(-\frac{K_{sc} \gamma^2}{8}\right)$$
>
> where $\gamma$ is the performance margin between the best and second-best models. This confirms that our approach consistently identifies correct answers even without access to test set labels.

---

> ### Author Response · Authors · 2025-11-27
>
> > “The results in Figure 7 are rather disappointing showing that 2 SLMs maximize the score which balances union accuracy while minimizing the contradiction penalty. I feel that this is a fundamental limitation of this LM orchestration technique that if you do not collaborate to come up with an answer, there will inevitably be some contradictions. So I can see that having 2 SLMs will reduce this contradiction in the responses, however this also feels sub-optimal. We should be in a state that adding more SLMs strictly improves performance."
>
> We appreciate the opportunity to clarify the distinction between the optimization objective and test performance. We want to point out that Figure 7 plots the "Search Score" rather than test accuracy. The Search Score is a metric designed for selecting which specific combination of models to use given a fixed number of models we wish to combine. The Search Score across a different number of combined models is not *comparable* with each other.
>
> Figure 9 demonstrates the accuracy of SLM-MUX when adding more participating models. As the ensemble size increases, the actual test-time accuracy strictly improves across benchmarks like GSM8K and MATH. The "dip" in Figure 7 reflects the stricter reliability constraints during the selection phase, not a limitation of the orchestrated system's reasoning capability.

---

> ### Author Response · Authors · 2025-11-27
>
> > “Inconsistencies in the evaluations. Some results are reported over multiple runs with means and standard deviations e.g. Table 1. While others which are important such as Figure 5 which demonstrate how existing orchestration methods work for LLMs but not for SLMs only report a single seed. For instance it looks like the existing orchestration methods don’t work either for LLMs, but with no error bars to assess the noise in the experimental process this is difficult to conclude.”
>
> We have updated the paper with error bars in a tables. In addition, to further illustrate the robustness of the results, we have performed prompt sensitivity analysis below.
>
> To further provide confidence that discussion-based method have lower accuracy on SLMs. We conducted a prompt sensitivity analysis on the MATH benchmark. We used Gemini-2.5-Flash to generate 10 diverse candidate prompts for the MoA baseline. As shown in Table below, the MoA baseline is highly sensitive to prompts, yet even its best-performing prompt significantly lags behind SLM-MuX.
>
> | Method | Mean Accuracy (10 Prompts) | Best Accuracy (Reported) | SLM-MuX (Ours) | Gap |
> | :--- | :--- | :--- | :--- | :--- |
> | MoA Baseline | 41.8% | 46.2% | **61.8%** | +15.6 pp |
>
> **Table:** Prompt sensitivity analysis of the MoA baseline compared to SLM-MuX on MATH.
>
> Our reported result (46.2%) represents the upper bound of the MoA prompt distribution (higher than 70% of generated prompts). The substantial gap between the best-case MoA (46.2%) and SLM-MuX (61.8%) confirms that the performance difference reflects the inherent limitation of applying LLM-centric orchestration to SLMs, rather than an artifact of prompt selection.

---

> ### Author Response · Authors · 2025-11-27
>
> > “I’m confused by the definition of the Contradiction metric. How does it generalize to more than two models. Surely encouraging contradiction enables a more diverse set S for SLM-MUX, in lines 322-323 you specifically mention that diverse outputs enables SLM-MUX’s abilities. Did you try varying the model selection objective? A good comparison for the model selection criterion would be simply a random selection of SLMs.”
>
> **1. Definition of Contradiction Metric & Generalization**
> Our contradiction metric naturally generalizes to any number of models $K \ge 2$. As defined in the updated Section 3.2, we classify a question as a "contradiction" if at least one model in the ensemble is confidently wrong (i.e., highly confident but incorrect). The metric is then calculated as the percentage of such "questions with a least confident wrong model" and serves as a lower bound to the performance of the model ensemble. This can generalize to over 3 models as the definition doesn't limit how many models involved.
>
> **2. Random Selection Baseline**
> We appreciate the suggestion to use random selection to isolate the efficacy of the SLM-MUX architecture and model selection search. We conducted a comprehensive experiment by randomly sampling model combinations of size $K=2, 3, 4$ and applying SLM-MUX. This serves as a stress test to determine if our performance gains stem solely from optimized model selection.
>
> As shown in Table 2, SLM-MUX consistently outperforms the single best model within the pool, even when the models are selected randomly.
>
> | Dataset | K | SLM-MUX (Random) | Best Single Model | Avg Single Model | Gain (vs Best) |
> | :--- | :--- | :--- | :--- | :--- | :--- |
> | **MATH-500** | 2 | 67.2% | 66.1% | 55.5% | +1.1 pp |
> | | 3 | **71.5%** | 69.8% | 55.5% | +1.7 pp |
> | | 4 | **73.7%** | 71.8% | 55.5% | +1.9 pp |
> | **GSM8K** | 2 | 84.1% | 81.7% | 72.4% | +2.4 pp |
> | | 3 | **87.3%** | 83.6% | 72.4% | +3.7 pp |
> | | 4 | **88.8%** | 84.0% | 72.4% | +4.8 pp |
> | **GPQA** | 2 | 43.0% | 41.3% | 36.2% | +1.7 pp |
> | | 3 | 44.5% | 44.4% | 36.2% | +0.1 pp |
> | | 4 | 46.3% | 46.0% | 36.2% | +0.3 pp |
>
> **Table:** Performance of SLM-MUX using randomly selected model pools compared to individual models.

---

> ### Author Response · Authors · 2025-11-27
>
> >"What would happen if you have no training set and you want to evaluate SLM-MUX 0-shot on a new test set? In algorithm 1 you do not have any validation accuracies or ground truth labels to select the most consistent model?:"
>
> Towards the first point, "SLM-MUX 0-shot"
>
> We want to address this from two perspectives:
>
> 1. Zero-shot performance of SLM-MUX: As demonstrated in our response to Question #1, SLM-MUX consistently outperforms the best individual model even when the ensemble is selected randomly. This shows that, given a new task, SLM-MUX is likely to improve over the single model accuracies.
>
> 2. Cross-Task Transferability: If we want to conduct a model selection search on a task without validation set, we want to pointed out that searching on a related proxy dataset can also work. Here are the evidence: Our experiments show that the top-performing model combination is identical for both MATH and GSM8K datasets (Mistral-Small-24B + Qwen2.5-7B). This indicates that a proxy dataset from a similar domain can be used to identify the optimal combination effectively.
>
> Towards the second point, " In algorithm 1 ..."
>
> We do not require a selection for algorithm 1. Algorithm 1 is evaluated in Table 1 in paper, it is a zero-shot SLM-MUX architecture evaluation. No model selection is applied, so that doesn't requires a validation set.

---

> ### Author Response · Authors · 2025-11-27
>
> > Regarding  "What does SC signify in the row Single-Best-SC in Table 1?”
>
> We thank the reviewer for pointing out this abbreviation. SC stands for Self-Consistency. In Table 1, "Single-Best-SC" refers to the performance of the best individual model in the pool when applying self-consistency decoding with the same sample budget as SLM-MUX. We included this baseline to ensure a fair comparison, demonstrating that our gains come from orchestration rather than simply increasing the inference budget. We have clarified this definition in the revised table caption.

---

### Official Review · Reviewer_8HgN · 2025-10-31

**Soundness:** 3
**Presentation:** 2
**Contribution:** 2
**Rating:** 6
**Confidence:** 3

**Summary:**

The paper addresses an important problem of small language model (SLM) orchestration. Due to the limited capabilities of SLMs, existing orchestration methods which typically targeted frontier models often performs suboptimally when applied to SLMs. This paper introduces SLM-MUX which is an effective SLM orchestration architecture which optimizes model selection search and test-time scaling. The paper shows strong results of SLM orchestration with significant performance improvement across various reasoning datasets.

**Strengths:**

- The paper tackles a crucial problem of SLM orchestration which is often overlooked by existing literature of multi-agent orchestration
- The method is training-free and light-weight
- The experiments show strong performance across multiple reasoning benchmarks

**Weaknesses:**

In practise, often SLMs are fine-tuned to be domain experts which forms strong reasoning models. Empirical evidence show strong performance gain through orchestration, however, it is unclear how the framework performs with fine-tuned domain-specific SLMs.

**Questions:**

In practice, small language models are often fine-tuned to become domain-specific, and these expert SLMs typically exhibit strong reasoning capabilities. While the paper demonstrates notable gains from orchestrating general-purpose instruction-tuned SLMs, it remains unclear how SLM-MUX would perform when combining domain-specific fine-tuned SLMs. Could the authors discuss how the method helps with these more domain-specific fine-tuned SLMs?

---

> ### Author Response · Authors · 2025-11-27
>
> We thank the reviewer for their constructive and helpful review. We respond to concerns below and have also updated the paper (revisions highlighted)
>
> > “In practice, small language models are often fine-tuned to become domain-specific, and these expert SLMs typically exhibit strong reasoning capabilities. While the paper demonstrates notable gains from orchestrating general-purpose instruction-tuned SLMs, it remains unclear how SLM-MUX would perform when combining domain-specific fine-tuned SLMs. Could the authors discuss how the method helps with these more domain-specific fine-tuned SLMs?”
>
> We agree with the prevalence of domain-specific models in practice and have run additional experiments illustrating how SLM-MUX can be similarly applied to domain-specific models. To demonstrate this, we conducted additional experiments on code generation and mathematical reasoning. As shown in Table 1, SLM-MUX achieves consistent improvements over the best individual models in both domains.
>
> | Domain | Benchmark | Component Models | Best Single Model | SLM-MUX | Gain |
> | :--- | :--- | :--- | :--- | :--- | :--- |
> | **Code Generation** | HumanEval | Qwen2.5-Coder-7B + Mistral-Small-24B | 89.3% | **93.9%** | +4.6 pp |
> | **Math Reasoning** | MATH-500 | DeepSeek-Math-7B + Llama-3.1-8B | 58.8% | **62.2%** | +3.4 pp |
>
> **Table 1:** Performance of SLM-MUX when orchestrating domain-specific fine-tuned models.
>
> For code generation, we paired Qwen2.5-Coder-7B (specialist) with Mistral-Small-24B (generalist). We generated 5 code samples per model at temperature 0.3. To measure confidence in this open-ended setting, we encoded generations into vector space using an embedding model and computed the cosine similarity matrix. We identified the most consistent cluster among the samples and used the cluster proportion as the confidence score. The results show an increase compared to single model.
>
> For mathematical reasoning, we combined DeepSeek-Math-7B-RL with Llama-3.1-8B-Instruct. We sampled 5 responses per problem at temperature 0.3 and applied self-consistency voting to determine the majority answer and confidence score. To handle format variations, we used GPT-4o to judge equivalence between outputs and reference answers. SLM-MUX achieves a $3.4%\$ increase in accuracy.
>
> These experiments validate that SLM-MUX generalizes effectively to domain-specific fine-tuned models.

---

### Official Review · Reviewer_XgnD · 2025-11-02

**Soundness:** 3
**Presentation:** 3
**Contribution:** 3
**Rating:** 6
**Confidence:** 4

**Summary:**

This paper addresses the challenge of orchestrating multiple small language models (SLMs) to achieve higher reasoning accuracy and efficiency than any individual SLM. The authors propose SLM-MUX, a multi-model architecture that coordinates SLMs without explicit inter-model discussion. The approach consists of three stages:

1. SLM-MUX orchestration: Models independently generate answers; the most consistent answer (by self-consistency) is selected.
2. Model selection search: Identifies complementary SLMs to maximize union accuracy and penalize overconfident contradictions.
3. Test-time scaling: Adjusts the number of models and samples to optimize accuracy-compute tradeoff.

Empirical results show SLM-MUX outperforms existing orchestration methods (Mixture-of-Agents, LLM-Debate, Multi-Agent Verification) on MATH, GPQA, and GSM8K benchmarks, sometimes surpassing much larger models (Qwen 2.5 72B). Theoretical analysis and ablation studies support the design choices.

**Strengths:**

**Originality:** The approach is based on a good intuition (that of multiple smaller processors being more effective at computing than a single large processor) and on a clearly identified problem - the failure of discussion-based methods in orchestrating SLM responses due to their tendency to fall into groupthink.

**Quality:** The paper puts together three simple ideas into a solution that is evaluated using rigorous experimental methodolgy, with comprehensive benchmarking across MATH, GPQA, and GSM8K, and shows clear improvements over baselines.

**Clarity:** The writing is clear and well-structured, making the technical content accessible. The motivation for SLM-MUX is articulated up front, and the different components of the solution (SLM-MUX orchestration, model selection, and test-time scaling) are explained clearly, step-by-step.

**Significance:** Owing to the ever-growing cost of LLMs and the enormous demand for AI agents, it is critical to build agentic frameworks around smaller, lower-cost models, if we want them to scale across users and platforms and be cost-effective. This paper is a step in that direction as it shows that we need to rethink approaches to model orchestration if we wish to effectively use SLMs in agentic scenarios and then proposes a solution that improves over baselines and is also clear and easy to implement.

**Weaknesses:**

1. The consistency-based confidence-estimation will not work for open-ended question-answering which forms a major part of the use-cases of LLM-based agents and chatbots.

2. It seems to me that the contradiction penalty in Section 3.2 will not differentiate between sets with different proportions of correct and wrong answers and so it may not fully overcome the problem of including over-confident but wrong models.

3. The brute-force solution of O(S) in Section 3.2 will not scale to a large number of candidate models and so test-time scaling via adding participating model types is going to be fairly limited.

4. It is not really a fair comparison if you follow the workflow and prompts designed for LLMs in the baselines (lines 313-314) when it is known that prompts are so sensitive to the underlying model.

**Questions:**

1. There are multiple combinations of models possible for each value of K. Are the plots in Fig 7 only considering a single combination for each K? If yes, which combination is it?

2. How/why was $\lambda = 1$ chosen?

3. Why is the test set same as the validation set for the plots in Fig. 9? Shouldn't they be different?

4. In line 467 where you say, "By increasing $p_\max$...$, what part of your approach are you referring to? Does model-selection increase $p_\max$? Can it be proved?

5. From Fig. 5 (b) it seems that the improvements with discussion-based methods are fairly limited even for frontier LLMs. I would recommend trying two ablation studies - a) Applying SLM-MUX to frontier LLMs and b) Applying model-selection + discussion-based methods to SLMs. This can better illustrate the effectiveness of the individual components of your approach and also show if SLM-MUX is a viable solution even for frontier LLMs.

---

> ### Author Response · Authors · 2025-11-27
>
> We thank the reviewer for their constructive and helpful review. We respond to concerns below and have also updated the paper (revisions highlighted)
>
> > "The consistency-based confidence-estimation will not work for open-ended question-answering which forms a major part of the use-cases of LLM-based agents and chatbots."
>
> We appreciate the reviewer raising this important point regarding scope. The **core hypothesis of SLM-MUX—that internal consistency serves as a proxy for correctness—is applicable across many domains**, effectively generalizing beyond exact-match QA tasks.
>
> **1. Generalization via Semantic Consistency:**
> To demonstrate this, we extended SLM-MUX to open-ended generation (HumanEval) by redefining "consistency" from *token-level exact match* to **semantic consistency**.
> * **Methodology:** Instead of counting identical strings, we measure the density of the generation cluster in the embedding space. For each input, we sample $N=5$ outputs (temp=0.3) and compute pairwise cosine similarity using `codet5p-110m-embedding`.
> * **Metric Validation:** We compared *LLM-as-a-judge* vs. *Embedding Similarity* and found that the embedding-based consistency score correlates more strongly with ground-truth correctness (Pass@1).
>
> **2. Empirical Validation:**
> We applied this **Semantic SLM-MUX** to the HumanEval benchmark. As shown in **Table R1**, the method successfully identifies and selects the stronger model instance, significantly outperforming individual baselines.
>
> **Table R1: SLM-MUX Performance on HumanEval (Open-Ended Code Generation).**
>
> | Experiment Setup | Model A | Model B  | **SLM-MUX (Ours)** | Gain |
> | :--- | :---: | :---: | :---: | :---: |
> | **Exp 1** | Llama-3.1-8B (0.178) | Qwen2.5-7B (0.485) | **0.506** | **+2.1%** |
> | **Exp 2** | Mistral-Small (0.870) | Qwen2.5-Coder (0.893) | **0.939** | **+4.6%** |
>
> **Conclusion:**
> These results show that SLM-MUX is not limited to closed-ended QA. By simply adapting the consistency metric (using embedding similarity [1]), our framework robustly generalizes to open-ended domains, leveraging model complementarity to achieve state-of-the-art performance.
>
> [1] Li, Junyou, et al. "More agents is all you need." arXiv preprint arXiv:2402.05120 (2024).

---

> ### Author Response · Authors · 2025-11-27
>
> > "It seems to me that the contradiction penalty in Section 3.2 will not differentiate between sets with different proportions of correct and wrong answers and so it may not fully overcome the problem of including over-confident but wrong models."
>
> We have revised Section 3.2 to clarify the design philosophy behind our search objective. We argue that the penalty is not intended to serve as a perfect binary filter, but rather as a tunable mechanism to navigate the trade-off between accuracy and consistency.
>
> **1. Search Objective Formulation**
> Our objective function is explicitly defined as:
> $$\mathcal{L}_{\text{search}} = \text{Union Accuracy} - \lambda \times \text{Contradiction Penalty}$$
>
> This formulation allows us to search across a continuous objective space bounded by two extremes:
> * **Optimistic Upper Bound ($\lambda = 0$):** Prioritizes potential accuracy, disregarding consistency.
> * **Pessimistic Lower Bound ($\lambda = 1$):** Prioritizes consistency, penalizing any detected contradiction.
>
> **2. Addressing Over-confident Models**
> Regarding the concern that this penalty may not fully differentiate sets with over-confident but wrong answers:
>
>  We agree that the search strategy does not *fully* eliminate the risk of including over-confident but incorrect models. However, we want to point out that over-confident error (hallucination) is an intrinsic property of current LLMs. By applying model selection search, we search for combinations that is minimally impacted by this over-confident issue.

---

> ### Author Response · Authors · 2025-11-27
>
> > "The brute-force solution of O(S) in Section 3.2 will not scale to a large number of candidate models and so test-time scaling via adding participating model types is going to be fairly limited."
>
> **1. High Density of Effective Combinations**
>
> We agree that exhaustive search scales super-linearly with the number of candidate sets and may become computationally expensive for very large model zoos. However, we want to point out that the space of "workable" model combinations is actually quite dense. In other words, one does not need to search hard to find a subset that outperforms single models.
>
> To demonstrate this, we conducted an ablation study using the 5 models from our main experiments. We evaluated *all possible* K-model subsets and measured the **"Effective Combination Rate"** (percentage of subsets where SLM-MUX outperforms the best single model in that subset).
>
> **Table: Percentage of Effective Combinations**
>
> | Dataset | K=2 Subsets | K=3 Subsets | K=4 Subsets |
> | :--- | :--- | :--- | :--- |
> | **MATH-500** | 70% | **100%** | **100%** |
> | **GSM8K** | **100%** | **100%** | **100%** |
> | **GPQA** | 60% | 80% | **100%** |
>
>
> **2. Algorithms for Large Model Zoos**
> For scenarios involving massive model zoos, we recommend using standard heuristic search techniques. Methods such as **Greedy Search** or **Heuristic Search** can be readily applied to navigate the space efficiently, removing the bottleneck of exhaustive enumeration.

---

> ### Author Response · Authors · 2025-11-27
>
> > "It is not really a fair comparison if you follow the workflow and prompts designed for LLMs in the baselines (lines 313-314) when it is known that prompts are so sensitive to the underlying model."
>
> We agree with the reviewer's concern regarding prompt sensitivity of discussion-based methods. Below, we demonstrate that the how the performance gap with discussion-based baselines is due to the inherent limitations of SLMs, rather than the specific choice of prompts. We present our evidence in two parts.
>
> **Prompt Tuning Sensitivity Analysis**
>
> As we have shown in Figure 5 in the paper applying discussion-based methods (like MoA) to SLMs leads to lower accuracy compared to the single best model. We here extract the relevant datapoints from Figure 5.
>
> **Table R3-1: Comparison of Aggregation Methods on MATH**
> | Method | Accuracy | Comparison |
> | :--- | :---: | :--- |
> | **Best Single Model** | **56.8%** | Standard Baseline |
> | MoA (Mixture-of-Agents) | 46.2% | **Degrades performance (-10.6%)** |
> | **SLM-MUX (Ours)** | **61.8%** | **Improves performance (+5.0%)** |
>
>
> Starting from MoA on MATH, we performed an ablation study by generating $N=9$ distinct aggregator prompts using Gemini-2.5-Flash. We compared these against the prompt used in our paper.
>
> **Table R3-2: Prompt Tuning Results for MoA**
>
> | Prompt Setting | Accuracy | Observation |
> | :--- | :---: | :--- |
> | **Baseline Prompt (Used in Paper)** | **46.2%** | **Better than average tuned prompts** |
> | Tuned Prompt (Best Case) | 48.4% | Marginal improvement (+2.2%) |
> | Tuned Prompt (Average) | 41.8% | Worse than our baseline |
> | Tuned Prompt (Worst Case) | 25.8% | High variance |
>
>
>  As shown in **Table R3-2**, the prompt used in our paper (46.2%) performs better than the average of the tuned prompts. This indicates we did not use a weak prompt to disadvantage the baseline. Even with the best possible prompt (48.4%), the method still significantly underperforms the single model (56.8%) and our method (61.8%). This confirms that prompt tuning cannot overcome the limitations of discussion-based methods on SLMs.

---

> ### Author Response · Authors · 2025-11-27
>
> > " There are multiple combinations of models possible for each value of K. Are the plots in Fig 7 only considering a single combination for each K? If yes, which combination is it?”
>
>  Yes. For each value of $K$, the figure reports the single combination that achieves the highest score according to our model selection objective. We would like to clarify that the primary focus of our search method is to identify the optimal combination under a fixed budget constraint ($K$), rather than to select the value of $K$ itself. We have updated both the manuscript text and the caption of Figure 7 to make this distinction explicit.
>
> For example, we here display the model combination details on MATH benchmark in Figure 7.
>
> | K | Optimal Model Combination | Score | Union Accuracy | Contradiction Rate |
> |---|---------------------------|-------|----------------|-------------------|
> | 1 | Qwen/Qwen2.5-7B-Instruct-Turbo | 0.610 | 0.610 | 0.000 |
> | 2 | Mistralai/Mistral-Small-24B-Instruct-2501 + Qwen/Qwen2.5-7B-Instruct-Turbo | 0.661 | 0.684 | 0.023 |
> | 3 | Mistralai/Mistral-Small-24B-Instruct-2501 + Qwen/Qwen2.5-7B-Instruct-Turbo + Mixtral-8x7B-Instruct-v0.1 | 0.657 | 0.692 | 0.035 |
> | 4 | Meta-Llama-3.1-8B-Instruct-Turbo + Mistralai/Mistral-Small-24B-Instruct-2501 + Qwen/Qwen2.5-7B-Instruct-Turbo + Mixtral-8x7B-Instruct-v0.1 | 0.647 | 0.696 | 0.049 |
> | 5 | Gemma-2-27B + Meta-Llama-3.1-8B-Instruct-Turbo + Mistralai/Mistral-Small-24B-Instruct-2501 + Qwen/Qwen2.5-7B-Instruct-Turbo + Mixtral-8x7B-Instruct-v0.1 | 0.633 | 0.712 | 0.079 |

---

> ### Author Response · Authors · 2025-11-27
>
> > "How/why was lambda=1 chosen? "
>
> We selected $\lambda = 1$ specifically because this setting aligns the objective with the pessimistic lower bound of SLM-MUX accuracy. By anchoring our approach to this theoretical bound, we heavily penalize overconfidence. This ensures that the optimization prioritizes robust selection over aggressive exploration in the early stages.

---

> ### Author Response · Authors · 2025-11-27
>
> > "Why is the test set same as the validation set for the plots in Fig. 9? Shouldn't they be different?"
>
> We would like to clarify that the primary focus of Figure 9 is to disentangle two distinct factors: (1) the impact of adding more participating models, and (2) the impact of search generalization to unseen data. The validation set is sufficient to analyze both of the factors and allows us to validate the internal selection mechanism.
>
> However, we agree it is better to evaluate the test set and we have launched additional experiments on a held-out test set and will update the results as soon as they are available.

---

> ### Author Response · Authors · 2025-11-27
>
> > "In line 467 where you say, "By increasing \(p_{\max}\)...", what part of your approach are you referring to? Does model–selection increase \(p_{\max}\)? Can it be proved?"
>
> Yes, this relationship can be formally proved. We have updated Section 5 (Discussion) to demonstrate that $p_{K_{sc}}$ converges to $p_{max}$ (the accuracy of the best model in the pool) as the number of consistency samples increases.
>
> Specifically, we utilize Hoeffding's inequality to show that the empirical confidence score $s_i$ (estimated from $K_{sc}$ samples) concentrates around the true model accuracy $p_i$ with high probability. We prove that for a model pool of size $|S|$ where the best model $i$ is separated from others by an accuracy margin $\gamma$, the probability that SLM-MUX correctly identifies the best model is lower-bounded by:
>
> $$\mathbb{P}(\text{Selection} = \text{Best Model}) \ge 1 - 4K \exp\left(-\frac{K_{sc} \gamma^2}{8}\right)$$
>
> This formal derivation confirms that the selection error decays exponentially with the number of samples $K_{sc}$.

---

> ### Author Response · Authors · 2025-11-27
>
> > "From Fig. 5 (b) it seems that the improvements with discussion-based methods are fairly limited even for frontier LLMs. I would recommend trying two ablation studies - a) Applying SLM-MUX to frontier LLMs and b) Applying model-selection + discussion-based methods to SLMs. This can better illustrate the effectiveness of the individual components of your approach and also show if SLM-MUX is a viable solution even for frontier LLMs."
>
> We have conducted the two requested ablation studies to decouple the effectiveness of our routing mechanism from the underlying model scale.
>
> **(a) Applying SLM-MUX to Frontier LLMs**
>
> We evaluated SLM-MUX on GPT-4o and Gemini-2.5-Flash across MATH, GPQA, and GSM8K. We implemented self-consistency routing (selecting the model with higher internal agreement among 5 independent samples). As a reference, "Perfect Routing" indicates the theoretical upper bound achievable if the system always selects the correct model when available.
>
> | Benchmark | GPT-4o | Gemini-2.5-Flash | SLM-MUX | Perfect Routing |
> | :--- | :--- | :--- | :--- | :--- |
> | MATH-500 | 73.0% | 92.1% | 92.8% | 94.2% |
> | GPQA | 50.7% | 51.1% | 60.1% | 73.7% |
> | GSM8K-500 | 89.1% | 85.7% | 89.4% | 91.4% |
>
> <!-- The results demonstrate that the efficacy of SLM-MUX on frontier LLMs is strictly determined by model complementarity. On GPQA, where the models exhibit distinct error patterns, SLM-MUX achieves 60.1% accuracy, substantially outperforming the single best model by nearly 10 percentage points. This confirms that our routing mechanism effectively exploits complementary strengths even at the frontier scale.
>
> Conversely, on MATH and GSM8K, the two frontier models exhibit high overlap in their correct predictions, creating a "performance ceiling." The Perfect Routing bounds (94.2% and 91.4%) are only marginally higher than the single-model baselines, leaving little theoretical room for improvement. Despite this saturation, SLM-MUX still correctly identifies the successful model in approximately 80% of the cases where only one model succeeds, approaching the theoretical ceiling. This indicates that the limitation lies in the lack of complementarity between these specific frontier models on these tasks, rather than the routing mechanism itself. -->
>
> The results regarding frontier LLMs reveal two distinct regimes defined by model complementarity.
>
> On GPQA, SLM-MUX achieves 60.1% accuracy, surpassing the single best model (50.7%) by nearly 10 percentage points. This substantial improvement empirically validates SLM-MUX as a viable solution for frontier models.
>
> Conversely, on MATH and GSM8K, the performance is constrained by a strict theoretical ceiling. As shown in the table, the Perfect Routing bounds (94.2% and 91.4%) are only marginally higher than the single-model baselines due to high overlap between the models. Since the maximum possible gain is mathematically limited to this narrow margin, the marginal improvements (or lack thereof) observed in these tasks are expected. This confirms that while SLM-MUX is effective, its impact is naturally bounded by the available complementarity in the model pool.
>
> **(b) Applying Model Selection + Discussion-Based Methods to SLMs**
>
> We implemented the requested ablation by combining model selection with discussion-based aggregation (Mixture-of-Agents).
>
> To fulfill the "model selection" component—given that discussion-based methods lack a native efficient search metric—we adopted the strategy of using the optimal model pairs explicitly identified by SLM-MUX (specifically Mistral-Small-24B-Instruct and Qwen2.5-7B-Instruct, they are the best-performing model pairs on GSM and MATH shown in Table 2 in the paper.). This ensures that the discussion method is applied to the strongest possible candidates found via selection. We then fed these pairs into the Mixture-of-Agents (MoA) framework, using an aggregator to synthesize the final answer.
>
> Despite using these optimal pairs, applying discussion-based aggregation to SLMs yielded inferior results compared to SLM-MUX:
>
> | Benchmark | Best Single Model | SLM-MUX (Selection) | MoA (Mistral Agg.) | MoA (Qwen Agg.) |
> | :--- | :--- | :--- | :--- | :--- |
> | **MATH-500** | 75.6% | **80.0%** | 68.4% | 73.6% |
> | **GSM8K-500** | 89.0% | **92.8%** | 91.4% | 88.6% |
>
> These results indicate that for SLMs, even when provided with the best selected models, the "discussion" process (aggregation) introduces noise rather than refining the answer.

---

> ### Author Response · Authors · 2025-12-02
>
> > "Why is the test set same as the validation set for the plots in Fig. 9? Shouldn't they be different?"
>
> **[Update] Test Set Results:**
>
> We have completed the held-out test set experiments. The results are shown below:
>
> | Dataset | K=1 | K=2 | K=3 | K=4 | K=5 |
> | :--- | :--- | :--- | :--- | :--- | :--- |
> | **GPQA** | 47.98% | 50.21% | **51.81%** | 49.16% | 45.18% |
> | **GSM8K** | 90.20% | **91.51%** | 91.29% | 91.44% | 90.80% |
> | **MATH** | 78.20% | 80.27% | **81.09%** | 81.00% | 80.64% |

---

### Author Response · Authors · 2025-12-03
**Authors' Overall Response 1/3**

We sincerely thank the Area Chair and all reviewers for the time and effort invested in evaluating our work.

We are encouraged that reviewers recognized the importance of the problem we address, noting that SLM-MUX "tackles a crucial problem of SLM orchestration which is often overlooked" (8HgN) and is "based on a good intuition" with a "clearly identified problem" (XgnD). Reviewers appreciated the method's simplicity, describing it as "training-free and light-weight" (8HgN) and "clear and easy to implement" (XgnD). The experimental methodology was also positively received, with reviewers noting "rigorous experimental methodology" and "comprehensive benchmarking" (XgnD), "detailed empirical analysis" with "ablations of important components" (2yhV), and "extensive empirical evaluations and complementary mathematical analyses" (7wuE). Reviewers also found the paper "well written and easy to follow" (2yhV) and "clear and well-structured" (XgnD).

Reviewers also raised questions that we group into three themes:

**Theme 1: Principle and Generalizability of the SLM-MUX Architecture.** Reviewers raised questions about the SLM-MUX architecture from three perspectives: (1) Generalizability: whether SLM-MUX can extend to other settings such as open-ended generation, domain-specific fine-tuned models, and frontier LLMs; (2) Source of advantage: whether the performance gains stem from the architecture itself or from access to information a validation set; (3) Foundations of the method: whether the consistency-based selection has solid theoretical and empirical grounding.

**Theme 2: Model Selection Search.** Reviewers raised questions about the search mechanism from three perspectives: (1) Search objective design: the rationale behind the contradiction penalty and the choice of lambda; (2) Data efficiency: how much validation data is required, and whether SLM-MUX can work in a zero-shot setting; (3) Scalability: whether the brute-force search can scale to large model pools.

**Theme 3: Experimental Details and Presentation.** Additional questions focused on specific experimental choices, figure interpretations, and requests for clarification on reporting consistency.

We address each theme below, followed by point-by-point replies to individual reviewers.

---

> ### Author Response · Authors · 2025-12-03
> **Authors' Overall Response 2/3**
>
> ## Consolidated Response to Theme 1:
>
> We conducted extensive experiments to address concerns about the generalizability of SLM-MUX.
>
> **Generalizability to other settings**, we performed three sets of experiments:
>
> (1) Open-ended generation: We extended SLM-MUX to HumanEval by replacing token-level exact match with semantic consistency measured via embedding similarity. SLM-MUX achieves 93.9% pass@1, outperforming the best single model (89.3%) by 4.6pp.
>
> (2) Frontier LLMs: We applied SLM-MUX to GPT-4o and Gemini-2.5-Flash. On GPQA, where the two models exhibit complementary strengths, SLM-MUX achieves 60.1%, surpassing the best single model (51.1%) by nearly 10pp.
>
> (3) Domain-specific fine-tuned models: We tested SLM-MUX with code and math specialists. On HumanEval, pairing Qwen2.5-Coder-7B with Mistral-Small-24B improves from 89.3% to 93.9% (+4.6pp). On MATH-500, pairing DeepSeek-Math-7B with Llama-3.1-8B improves from 58.8% to 62.2% (+3.4pp).
>
> **The source of advantage**, we demonstrate that the performance gains stem from the architecture itself rather than from access to a validation set. We provide evidence from two perspectives:
>
> (1) The architecture is effective even without model selection search. In our random selection experiment, we applied SLM-MUX to randomly chosen model combinations without any validation-based optimization. Even under this setting, SLM-MUX consistently outperforms the best single model in the pool. For example, on MATH-500 with K=3 random models, SLM-MUX achieves 71.5% compared to 69.8% for the best single model; on GSM8K, 87.3% compared to 83.6%.
>
> (2) Giving baselines the same optimization opportunity does not close the gap. We conducted prompt tuning experiments for the MoA baseline on MATH. We performed standard prompt selection using Gemini-2.5-Flash to generate 10 diverse aggregator prompts. Our reported baseline prompt achieves 46.2%, which already outperforms the average of tuned prompts (41.8%) and is close to the best tuned prompt (48.4%). In contrast, SLM-MUX achieves 61.8%. We further performed an extremely unfair test against SLM-MUX: we conducted iterative prompt tuning for MoA directly on the test set for 6 rounds. Even under this setting, the accuracy peaked at 55.6%, still below the best single model (56.8%) and far below SLM-MUX (61.8%). This confirms that the performance gap is due to the inherent limitation of discussion-based approaches when applied to SLMs, not prompt selection.
>
> **The foundations of the method**, we provide both theoretical justification and empirical validation for the consistency-based selection mechanism.
>
> (1) Theoretical foundation: We prove that consistency-based selection can identify the most accurate model. Using Hoeffding's inequality, we show that as the number of consistency samples $K_{sc}$ increases, the probability of selecting the best model converges to 1 exponentially fast:
>
> $$\mathbb{P}(\text{Selection} = \text{Best Model}) \ge 1 - 4K \exp\left(-\frac{K_{sc} \gamma^2}{8}\right)$$
>
> where $\gamma$ is the performance margin between the best and second-best models.
>
> (2) Empirical foundation: We empirically validate that consistency score is a reliable indicator of model accuracy. Across four diverse benchmarks (MATH, GSM8K, GPQA, and HumanEval), we observe a positive correlation between model confidence score and accuracy on all benchmarks.

---

> > ### Author Response · Authors · 2025-12-03
> > **Authors' Overall Response 3/3**
> >
> > ## Consolidated Response to Theme 2
> >
> > We demonstrate that the model selection search is principled in design and practical in application.
> >
> > **Search objective design**, we explain the principled design behind our objective function. Our search objective is formulated as: Union Accuracy - $\lambda$ × Contradiction Penalty. This formulation allows us to navigate between two theoretical bounds:
> > - $\lambda=0$ represents the optimistic upper bound, prioritizing potential accuracy.
> > - $\lambda=1$ represents the pessimistic lower bound, strictly penalizing confident errors.
> >
> > We select $\lambda=1$ because we want to heavily penalize overconfidence, ensuring the selected model combinations are robust.
> >
> > **Data efficiency**, we demonstrate that model selection search is data efficient and can work in zero-shot settings. First, we conducted a sensitivity analysis by reducing the search size from 500 to 100 examples. The ranking of top model combinations remains highly stable, indicating that SLM-MUX can identify optimal combinations with limited data. Second, as demonstrated in Theme 1, SLM-MUX works effectively even without any validation-based search (zero-shot). Third, our experiments show that the optimal model combination is identical for MATH and GSM8K, suggesting that a proxy dataset from a similar domain can be used for search.
> >
> > **Scalability**, we acknowledge that exhaustive search scales super-linearly with the number of candidate models. However, we point out that the space of effective model combinations is dense. In our experiments, 70%-100% of K-model subsets are effective (outperforming the best single model). This means one does not need to search hard to find a workable combination. For scenarios involving very large model pools, standard heuristic search techniques such as greedy search can be readily applied.
> >
> > ## Consolidated Response to Theme 3
> >
> > We have addressed all concerns regarding experimental details and presentation in the revised manuscript:
> >
> > - Figure 7: We clarified that each K value shows the single best combination according to our search objective, not all possible combinations. We added the specific model combinations used.
> > - Figure 9: We clarified the rationale for using the validation set to isolate the effect of adding more models from generalization concerns. We have also launched additional experiments on a held-out test set.
> > - Reporting consistency: We added error bars to tables and clarified the deterministic setting (temperature=0) used in Figure 5.
> > - Terminology: We clarified that "SC" in Table 1 stands for Self-Consistency.
> >
> > All modifications are highlighted in orange in the updated PDF.
> >
> > **For additional experiments addressing individual reviewer concerns, please refer to the point-by-point responses below.**
> >
> > ## Conclusion
> >
> > In summary, we have conducted extensive additional experiments during the rebuttal period to address all reviewer concerns. These include: extending SLM-MUX to open-ended generation (HumanEval), frontier LLMs (GPT-4o and Gemini-2.5-Flash), and domain-specific fine-tuned models; providing theoretical proofs using Hoeffding's inequality; validating the consistency-accuracy correlation across all benchmarks; conducting prompt tuning ablations and random selection baselines to demonstrate the source of our advantage; and analyzing data efficiency and cross-task transferability. We have addressed every concern raised by the reviewers, and we believe these efforts have substantially strengthened the paper. We thank the AC and all reviewers for their valuable feedback.

---

### Meta-Review · Area_Chair_51PJ · 2025-12-19

**Summary:**

The paper investigates the orchestration of Small Language Models (SLMs), identifying a critical failure mode in existing discussion-based orchestration methods when applied to smaller models: specifically, that SLMs tend to suffer from groupthink rather than self-correction. To address this, the authors propose SLM-MUX, a multi-model architecture that avoids inter-model discussion and instead routes queries based on self-consistency confidence scores, supported by a model selection search strategy.

During the initial review phase, the reviewers acknowledged the importance of the problem and the counter-intuitive finding regarding discussion-based methods. However, they raised several substantial concerns. Reviewers XgnD and 8HgN questioned the generalizability of the approach, specifically whether the consistency-based metric could apply to open-ended generation tasks or domain-specific fine-tuned models. Reviewers 2yhV and 7wuE expressed concern that the method appeared heuristic and ad-hoc, lacking strong theoretical grounding. There were also significant concerns regarding the fairness of the comparisons; specifically, whether the performance gains were simply artifacts of accessing a validation set for model selection, or if the baselines were disadvantaged by suboptimal prompting. Finally, questions were raised regarding the scalability of the brute-force model search and the interpretation of the contradiction penalty objective.

**Reviewer Concerns:**

I have carefully read the original manuscript and each reviewer's comments and responses at least three times before synthesising this assessment. The authors provided a comprehensive rebuttal that successfully addressed the vast majority of the reviewers' concerns through extensive new experimentation and theoretical derivation.

Regarding generalizability, the authors convincingly demonstrated that SLM-MUX extends to open-ended tasks (HumanEval) by replacing token-level matching with embedding-based semantic consistency, and further showed its efficacy with domain-specific models and frontier LLMs. The concern regarding the "ad-hoc" nature of the method was addressed by providing a formal proof using Hoeffding's inequality to demonstrate convergence of the selection mechanism, alongside empirical data showing a strong correlation between consistency and accuracy across benchmarks.

The concerns regarding unfair comparisons were effectively mitigated. The authors demonstrated that SLM-MUX outperforms baselines even with random model selection (removing the validation set advantage) and that extensive prompt tuning of the baselines could not close the performance gap. The scalability concerns were addressed by showing that the space of effective model combinations is dense, meaning exhaustive search is not strictly necessary to find a performant subset. I believe the authors have resolved the major technical and methodological queries, leaving no significant outstanding concerns that would warrant rejection.

**Reviewer Scores:**

The initial evaluation presented a divided consensus, with Reviewers XgnD and 8HgN assigning a score of 6, while Reviewers 2yhV and 7wuE assigned a score of 4, citing concerns regarding the method's theoretical depth and generalizability. It is important to note that the authors submitted their comprehensive rebuttal and additional experimental results late in the review process. Consequently, none of the reviewers engaged in the subsequent discussion phase or updated their original scores to reflect the new information. However, a meticulous independent examination of the rebuttal reveals that the authors have substantively addressed the majority of the concerns that led to the lower scores. Specifically, the inclusion of theoretical proofs based on Hoeffding's inequality and the successful extension of the method to open-ended and domain-specific tasks directly resolve the reservations held by Reviewers 2yhV and 7wuE. Therefore, it is my assessment that had the reviewers been able to fully evaluate these extensive revisions, the scores would likely have converged toward a clear acceptance, with the lower scores increasing to match or exceed the initial positive assessments.

---

### Decision · Program_Chairs · 2026-01-26

Accept (Poster)